# Answering Counterfactual Queries on Graph Databases

## Abstract

Counterfactual analysis on graph data is central to causal reasoning and interpretability, yet existing graph-based methods rely on ad hoc perturbations and remain tied to model behavior rather than underlying data. To address this challenge, we introduce **Counterfactual Graph Database (CF-GDB) queries**, the first query-based framework for counterfactual reasoning on graphs that grounds counterfactuals in verifiable database instances. Our approach abstracts graphs into semantically meaningful concepts and compares them using a hypergraph-based distance that integrates local structure with global semantics. To ensure efficiency and scalability, we propose two complementary indices: the *Concept Distribution Index (CDI)*, a histogram that provides certified lower bounds, and the *Concept Semantic Index (CSI)*, a continuous embedding that provides upper bounds. These indices yield provably tight sandwich guarantees and enable efficient candidate pruning while preserving the fidelity of counterfactual retrieval. Using 8 read data sets across 4 domains, CF-GDB improves accuracy by over 20% and achieves up to 20× faster performance, demonstrating both fidelity and scalability.

## 1 Introduction

Counterfactual analysis (Rubin, 1974) has found broad applications in machine learning (Wachter et al., 2017) and data science (Karimi et al., 2021). Unlike factual explanations that justify observed outcomes, counterfactuals identify minimal yet semantically valid perturbations sufficient to alter a model's prediction (Wachter et al., 2017; Mothilal et al., 2020). Such perturbations correspond to actionable alternatives, such as treatment adjustments in healthcare (Zhang et al., 2025), portfolio rebalancing in finance (Zhang et al., 2025), or precedent revisions in law (Zhang et al., 2023), providing evidence that is both interpretable and verifiable. Moreover, counterfactuals improve generalization (Tan et al., 2022), enhance robustness (Bajaj et al., 2021), and are indispensable in high-stakes domains where explanations must translate into practical guidance.

Recent work has explored counterfactual analysis in graph learning (Ma et al., 2022; Verma et al., 2024b; He et al., 2024; Fournier & Medya, 2025; Giorgi et al., 2025) by perturbing nodes, edges, or attributes to generate alternative graphs. However, these methods primarily focus on *model behavior* rather than *data-grounded evidence*, and face three key limitations: (i) **model dependence** on gradients or weights (Bajaj et al., 2021; Armgaan et al., 2024; Lucic et al., 2022), which confines them to white-box settings and renders counterfactuals unstable in practice (McCoy et al., 2022; Shu et al., 2024); (ii) **domain validity**, as arbitrary edits often violate structural constraints, leading to, for example, invalid molecules (Zhang et al., 2025) or unrealistic transactions (Gan et al., 2021); and (iii) **limited transferability**, since instance-specific edits lack grounding in database schemas, making them difficult to reuse across queries or datasets (Giorgi et al., 2025).

In this paper, as our first contribution, we propose the **Counterfactual Graph Database (CF-GDB)**, a novel framework that reframes counterfactual reasoning as a query problem over graph databases. Unlike prior approaches that generate perturbed graphs to flip model predictions (Ying et al., 2019; Lucic et al., 2022), CF-GDB retrieves dataset-grounded,

domain-valid counterfactuals, ensuring explanations are anchored in verifiable instances—an essential requirement in high-stakes domains such as law (Zhang et al., 2023), finance (Gan et al., 2021), and healthcare (McCoy et al., 2022). This task is challenging because counterfactuals involve coordinated changes to node features, edges, and topology, where even small edits can fundamentally alter semantics (e.g., substituting a carbon atom with nitrogen transforms a benzene ring into a pyridine ring, changing its toxicity (Zhang et al., 2025)). Traditional similarity measures, such as graph edit distance (Gao et al., 2010) or whole-graph embeddings (Bai et al., 2019; Zheng et al., 2014), are either too fine-grained or too coarse, and are further constrained by Weisfeiler–Leman expressivity (Xu et al., 2019; Morris et al., 2023), offering limited guidance for localizing meaningful counterfactual differences (Armgaan et al., 2024; Giorgi et al., 2025).

To implement CF-GDB efficiently, as our second contribution, we introduce **Concept-Based Counterfactual Graph Query ($C^2$GQ)**, which reframes counterfactual reasoning as a query problem in a shared **graph concept space**. Graph concepts serve as semantic prototypes that cluster structurally and semantically similar subgraphs across a database of graphs (e.g., rings in molecules, communities in networks), enabling application-aware counterfactual reasoning beyond naive edge and vertex edits. Each graph is represented as a distribution over concepts, and differences are measured by a **hypergraph-based concept distance** grounded in unbalanced optimal transport (Vayer et al., 2020). This distance jointly captures fine-grained local changes (e.g., removing a molecular ring) and global distributional shifts (e.g., reconfiguring community structures), ensuring interpretability and domain validity across multiple scales.

To achieve scalable counterfactual queries, as the third contribution, we introduce two indices inspired by graph homomorphism theory (Dell et al., 2017): the **Concept Distribution Index (CDI)**, a histogram that aggregates concept counts to provide certified lower bounds, and the **Concept Semantic Index (CSI)**, a continuous embedding that encodes concept semantics to yield upper bounds. Beyond efficiency, our framework offers guarantees absent in prior work, including query success rates, provably correct pairwise rankings, and safe pruning rules that never discard valid counterfactuals. As a whole, these results establish the first theoretical foundation for counterfactual queries on graphs, in contrast to previous heuristic perturbation methods. Experiments using 8 real data sets across 4 domains demonstrate that $C^2$GQ improves accuracy by over 20% with 20× faster queries.

## 2 Related Work

**Counterfactual Analysis.** Counterfactual analysis explains model predictions by identifying minimal input changes that alter outcomes (Wachter et al., 2017), and has been widely applied in high-stakes domains (Zhang et al., 2025; 2023). While effective for tabular and textual data (Pawelczyk et al., 2020; Yang et al., 2021), graphs pose unique challenges because local edits can propagate globally (Ying et al., 2019). Early graph-based approaches (Ying et al., 2019; Cai et al., 2025; Lucic et al., 2022; Tan et al., 2022) primarily used mask-based deletions to highlight influential substructures (Prado-Romero et al., 2024), but these methods are closer to model explanations than true counterfactuals (Wachter et al., 2017). Subsequent work has explored generative models (Ma et al., 2022), heuristic edits (Abrate & Bonchi, 2021; Bajaj et al., 2021), and bi-level optimization (Giorgi et al., 2025), followed more recently by motif-based (He et al., 2024), contrastive (Fournier & Medya, 2025), and RL-driven strategies (Verma et al., 2024b). However, these methods largely focus on single-graph perturbations to flip model predictions, and thus suffer from strong model dependence, weak domain validity, and limited transferability. In contrast, we are the first to formalize query-based counterfactual analysis on graphs, enabling retrieval of counterfactuals with semantically meaningful concept differences from databases. Our framework ensures domain validity by leveraging graphs in the input database, introduces concept-based indices for scalability and fidelity, and provides certified bounds on query correctness, bridging local explanations with database-grounded reasoning.

**Graph Database.** Graph queries (Zhao & Han, 2010) provide principled tools for retrieval and analysis, for subgraph matching (Sun & Luo, 2020), reachability (Castanón et al., 2015), shortest-path search (Sommer, 2014), and motif discovery (Lin et al., 2016),

with applications in many domains, such as social networks (Angles et al., 2013), recommendation (Ren et al., 2017), and knowledge graphs (Hogan et al., 2021). To scale these tasks, graph databases such as Neo4j (Robinson et al., 2015), Titan (DB-Engines, 2020), and Geabase (Fu et al., 2019) integrate native storage, indexing, and query languages. Graph learning (Kipf & Welling, 2017) further advanced this line of work by providing neural representations, enabling tighter integration with databases for downstream tasks (Walke et al., 2024; Zhou et al., 2023) through platforms such as GraphScope (Fan et al., 2021), AliGraph (Yang, 2019), DGL (Zheng et al., 2020), and neural graph databases (Besta et al., 2022). However, most graph learning databases prioritize scalability for model development or deployment. In contrast, our **Counterfactual Graph Database (CF-GDB)** shifts the focus from prediction accuracy to interpretability by querying semantic counterfactuals that flip outcomes. Instead of embedding graphs solely for prediction, CF-GDB grounds counterfactual queries in real database instances and reframes counterfactual analysis as a retrieval problem, bridging scalable database support with explainable reasoning.

## 3 Problem Formulation

Consider a graph $G = (V_G, E_G, \mathbf{X}_G)$, where $V_G$ is the set of nodes, $E_G \subseteq V_G \times V_G$ the edges, and $\mathbf{X}_G \in \mathbb{R}^{|V_G| \times d}$ the node-feature matrix, with each row $\mathbf{x}_v \in \mathbb{R}^d$ representing the feature vector of node $v \in V_G$. The parameter $d$ denotes the feature dimensionality. Each graph is associated with a class label $y \in \mathcal{Y}$, where $\mathcal{Y}$ is a finite label set. A **graph database** is defined as $\mathcal{D} = (G_i, y_i) \mid 1 \leq i \leq n, , y_i \in \mathcal{Y}$, a collection of labeled graphs. A complete notation table is provided in Table 3 of Appendix A.

**Problem.** Given a graph database $\mathcal{D}$ and a query $Q = (G, y)$, where $G$ is a graph and $y \in \mathcal{Y}$ its label, the *counterfactual* of $Q$ with respect to $\mathcal{D}$ is defined as $(\tilde{G}^*, \tilde{y}^*) = \arg\min_{(\tilde{G}, \tilde{y}) \in \mathcal{D}, \tilde{y} \neq y} \Delta(G, \tilde{G})$, where $\Delta(G, \tilde{G}) \geq 0$ denotes a distance measure between graphs $G$ and $\tilde{G}$. Note that the query graph $Q$ does not have to belong to $\mathcal{D}$. A **counterfactual graph query (CGQ)** is the task of retrieving such a counterfactual from the database.

Defining an effective distance measure is critical for counterfactual graph queries. A natural choice is edit-based metrics, but these are computationally expensive, as they typically require both graph alignment (e.g., subgraph isomorphism or homomorphism, which are #P-hard (Dell et al., 2017)) and subsequent distance computation (e.g., graph edit distance, NP-hard (Gao et al., 2010)). More importantly, naive node-by-node alignment is semantically uninformative, as it fails to capture structural patterns shared across graphs (Armgaan et al., 2024). For example, in molecular graphs, replacing a carbon atom with nitrogen is not merely a single-atom substitution (Ma et al., 2022); it transforms a benzene ring into a pyridine ring, thereby altering key molecular properties such as toxicity or reactivity. The true unit of change is thus a higher-level structural element, such as a *functional group*, which naturally serves as a graph concept. Similarly, in social network classification, removing a single friendship edge rarely affects the predicted label. What matters instead is the presence or absence of larger structures, such as *communities* (Bajaj et al., 2021), which serve as graph concepts that distinguish between "fragmented" and "cohesive" networks.

These limitations highlight the need for a higher-level abstraction that enables semantically meaningful comparison and retrieval. We introduce **graph concepts** (or simply concepts), defined as probabilistic clusters of nodes and relations forming coherent semantic units (e.g., communities in social networks or functional groups in molecules). Graph concepts act as reusable *principal components* across graphs and serve as atomic units for counterfactual editing and retrieval. Identifying such concepts for counterfactual queries is nontrivial: they must be learned directly from the database and capture semantic differences between graphs that drive label changes. This requires both a principled method for concept learning and a mechanism to leverage them in queries. Unlike supervised approaches such as Concept Bottleneck Models (Koh et al., 2020; Barbiero et al., 2024), which rely on human-defined labels as auxiliary targets, our approach ($C^2GQ$) automatically learns and extracts concepts for counterfactual queries without external supervision.

# 4 CONCEPT-BASED COUNTERFACTUAL GRAPH QUERY

To overcome the limitations of existing approaches—either computationally prohibitive and overly localized (edit-based) or semantically coarse and unable to capture fine-grained counterfactual differences (embedding-based)—we propose **Concept-Based Counterfactual Graph Query ($C^2GQ$)**. Unlike prior methods tied to individual instances, $C^2GQ$ mines graph concepts from multiple graphs in a database, uncovering recurring structures (e.g., functional groups, communities, motifs) that serve as prototypes for capturing differences that drive prediction flips. By introducing a **hypergraph-based concept distance**, $C^2GQ$ enables multi-scale, domain-valid comparisons, yielding a more expressive metric for counterfactual queries. Building on this foundation, the **Concept Distribution Index (CDI)** and **Concept Semantic Index (CSI)** provide certified candidate-quality bounds by approximating $\Delta(G, \tilde{G})$ with lower and upper guarantees. These indices ensure that index-based search delivers model- and task-independent validity, certifies retrieval rankings, and safely prunes irrelevant candidates (Section 5). Taken as a whole, these guarantees establish $C^2GQ$ as a new paradigm for scalable and interpretable graph counterfactual queries in databases. Detailed pseudocode is provided in Appendix A.4.

## 4.1 CONCEPT EXTRACTION VIA STRUCTURAL EMBEDDINGS

A central challenge in counterfactual graph queries is to find a representation that captures meaningful structures while remaining scalable. Prior approaches often fall into two extremes: node-level representations, which are overly fine-grained and sensitive to small perturbations (Gao et al., 2010; Ying et al., 2019; Ma et al., 2022), and graph-level embeddings, which scale well but collapse local variations and overlook higher-order substructures (Bai et al., 2019; Wang et al., 2024; Li et al., 2025; Abrate & Bonchi, 2021). To bridge this gap, we introduce **graph concepts** as intermediate representations. Concepts serve as recurring prototypes (e.g., functional groups, communities) that act as stable yet sensitive anchors across instances, enabling large-scale database operations.

To effectively extract concepts, we employ Graph Neural Networks (GNNs) (Kipf & Welling, 2017; Veličković et al., 2018) to obtain node embeddings $\mathbf{h}_v$, and denote their collection by $\mathcal{H} = \{\mathbf{h}_v \mid v \in V_G, G \in \mathcal{D}\}$. Importantly, GNN-derived embeddings are structure-aware and semantically aligned across graphs (Armgaan et al., 2024), and thus capture higher-order patterns beyond raw features, crucial for reliable and transferable concept extraction (see Appendix A.1). We then cluster $\mathcal{H}$ into $K$ prototypes via $K$-means:

$$\min_{\{a(v)\}, \{\mathbf{c}_k\}_{k=1}^K} \sum_{\mathbf{h}_v \in \mathcal{H}} \|\mathbf{h}_v - \mathbf{c}_{a(v)}\|_2^2, \tag{1}$$

where $a : \mathcal{H} \to [K]$ assigns each embedding to a cluster index, and the corresponding centroid is $\mathbf{c}_{a(\mathbf{h}_v)} \in \mathbf{C}$. Each centroid $\mathbf{c}_k$ serves as a *concept prototype*, representing a recurring structural pattern across the database. The collection of prototypes $\mathbf{C}$ forms a global semantic dictionary that (i) amortizes computation by shifting clustering offline, and (ii) provides a shared basis that renders distance measures both efficient and semantically meaningful for counterfactual graph queries.

## 4.2 HYPERGRAPH-BASED CONCEPT DISTANCE

With concepts as the representation basis, the next challenge is to define a principled distance for comparing and mining multiple database graphs in concept space. Existing measures fall short: edit-based metrics (Gao et al., 2010; Chang et al., 2017) are computationally prohibitive, while embedding similarities (Bai et al., 2019; Zheng et al., 2014) obscure fine-grained counterfactual edits. In contrast, Optimal Transport (OT) (Petric Maretic et al., 2019) offers a more general framework, as it preserves structural correspondences, accommodates imbalance, and naturally models both discrete edits (via mass movement) and semantic shifts (via distances in embedding space). Yet node-level OT (Yu et al., 2025) remains insufficient for counterfactual queries, as it overlooks higher-order dependencies and cannot capture concept-level transformations such as ring substitutions in molecules or community rewiring in social networks. To address these limitations, we introduce the

**hypergraph-based concept distance**, which extends unbalanced OT (Vayer et al., 2020) to jointly model discrepancies at both node and concept levels across multiple graphs—a capability not achieved by prior approaches (Lucic et al., 2022; Abrate & Bonchi, 2021; Bajaj et al., 2021)—thereby providing a more expressive metric for counterfactual reasoning.

Each graph $G$ is mapped to a hypergraph $H_G = (V_G, F_G)$, where each factor $f \in F_G$ groups nodes with the same concept assignment $a(v)$. This reflects the tendency of nearby nodes to share concepts, as well as the alignment of actors exhibiting similar structural patterns in the graph. Factor embeddings are then computed as $\mathbf{g}_f^G = \psi(\mathbf{h}_v^G : v \in f)$, where $\psi$ is a permutation-invariant operator such as mean, sum, or attention (Xu et al., 2019; Lee et al., 2019). In this way, factor representations capture higher-order dependencies—such as communities in social networks or functional groups in molecules—that cannot be expressed solely through pairwise edges (see Appendix A.2 for details).

For two graphs $G$ and $\tilde{G}$, we define a block-diagonal transport plan

$$T = \begin{bmatrix} T^{V_G V_{\tilde{G}}} & 0 \\ 0 & T^{F_G F_{\tilde{G}}} \end{bmatrix}, \qquad T^{V_G V_{\tilde{G}}}, T^{F_G F_{\tilde{G}}} \geq 0, \tag{2}$$

where each row of $T^{V_G V_{\tilde{G}}}$ (resp. $T^{F_G F_{\tilde{G}}}$) corresponds to a node $v \in V_G$ and each column to a node $\tilde{v} \in V_{\tilde{G}}$, with $T_{v,\tilde{v}}^{V_G V_{\tilde{G}}}$ denoting the transported mass aligning $v$ with $\tilde{v}$. Standard partial matching constraints (Chapel et al., 2020) enforce valid alignments, disentangling node- and concept-level distances, and yielding interpretable edits such as removals or substitutions.

By solving the transportation plan, the hypergraph-based distance is formally defined as

$$\Delta(G, \tilde{G}) = \min_{T^{V_G V_{\tilde{G}}}, T^{F_G F_{\tilde{G}}} \geq 0} \lambda_{\text{sub}}^V \sum_{v,v'} T_{vv'}^{V_G V_{\tilde{G}}} \delta_V(\mathbf{h}_v^G, \mathbf{h}_{v'}^{\tilde{G}}) + \lambda_{\text{del}}^V \Big( |V_G| - \sum_{v,v'} T_{vv'}^{V_G V_{\tilde{G}}} \Big) + \lambda_{\text{ins}}^V \Big( |V_{\tilde{G}}| - \sum_{v,v'} T_{vv'}^{V_G V_{\tilde{G}}} \Big)$$
$$+ \lambda_{\text{sub}}^F \sum_{f,f'} T_{ff'}^{F_G F_{\tilde{G}}} \delta_F(\mathbf{g}_f^G, \mathbf{g}_{f'}^{\tilde{G}}) + \lambda_{\text{del}}^F \Big( |F_G| - \sum_{f,f'} T_{ff'}^{F_G F_{\tilde{G}}} \Big) + \lambda_{\text{ins}}^F \Big( |F_{\tilde{G}}| - \sum_{f,f'} T_{ff'}^{F_G F_{\tilde{G}}} \Big). \tag{3}$$

Here, $\lambda_{\text{sub}}^*, \lambda_{\text{del}}^*, \lambda_{\text{ins}}^* \geq 0$ denote the penalties for substitution, deletion, and insertion at both node and factor levels, while $\delta_V$ and $\delta_F$ are squared $\ell_2$ costs (Bai et al., 2019; Dixit et al., 2022). These weights can be estimated from data by applying controlled perturbations and fitting regression models that map histogram shifts to unit costs.

Notably, the hypergraph-based distance provides flexible control over the granularity of concept comparison: setting $F = \varnothing$ recovers the classical graph edit distance (Gao et al., 2010); choosing $F = V$ with a single factor yields whole-graph embedding similarity (Bai et al., 2019); and defining local neighborhoods as factors enables structure-aware alignment through graph coarsening (Ying et al., 2018; Lee et al., 2019). By aligning concepts at appropriate granularities, the distance faithfully captures the minimal edits that drive label flips. Moreover, domain-specific choices of $F$ enforce validity under real-world constraints, such as valence preservation in molecules (Gilmer et al., 2017), regulatory plausibility in finance (Rao et al., 2021), and spatio-temporal smoothness in traffic (Li et al., 2018).

### 4.3 Graph-Level Indices from Concepts

Although the hypergraph-based concept distance is expressive, its direct application at scale is computationally prohibitive. To address this, we introduce two lightweight graph-level indices: (i) the **Concept Distribution Index (CDI)**, a histogram $\boldsymbol{\sigma}_G$ with $\sigma_G(k) = \sum_{v \in V_G} \mathbf{1}(a(v) = k)$, which provides a bag-of-concepts view and reflects discrete edits such as insertions or deletions; and (ii) the **Concept Semantic Index (CSI)**, a dense embedding $\mathbf{z}_G = \sum_{k=1}^K \frac{\sigma_G(k)}{|V_G|} \mathbf{c}_k$ that aggregates across multiple graphs to capture concept-level semantics. Theoretically, CDI tracks homomorphism counts to yield a certified lower bound, while CSI encodes these counts in continuous space to provide an upper bound. Together, they sandwich the hypergraph-based concept distance $\Delta(G, \tilde{G})$ (Theorems 1–6), offering the first certified guarantees for scalable and faithful counterfactual queries—unlike prior single-graph methods that rely on heuristic perturbations without guarantees (Ying et al., 2019; Abrate & Bonchi, 2021; Ma et al., 2022) (detailed Appendix A.3).

Finally, we introduce a three-step method for efficient counterfactual graph query answering:

1. **Label hashing.** Partition the database into buckets by label, and restrict the search to buckets with labels different from the query label $y$, i.e., $\mathcal{B}_{\neg y}$.
2. **Dual pruning.** Within each bucket, first prune candidates using the $\ell_1$ histogram distance $\delta_{\text{freq}}(G, \tilde{G}) = \|\boldsymbol{\sigma}_G - \boldsymbol{\sigma}_{\tilde{G}}\|_1$ (retaining the $\alpha M$ closest), and then refine with the $\ell_2$ semantic distance $\delta_{\text{sem}}(G, \tilde{G}) = \|\mathbf{z}_G - \mathbf{z}_{\tilde{G}}\|_2$ to select the top-$M$ candidates.
3. **Fine re-ranking.** Apply the hypergraph-based distance to the pruned set $\mathcal{D}_{\text{pruned}}$, yielding interpretable edits $\Delta(G, \tilde{G})$ from the optimal transport plan $T^\star$.

To further accelerate counterfactual queries, steps (1) and (2) are precomputed offline: label partitioning is stored in a hash map, and both indices are organized in a KD-tree. At query time, only step (3) is executed on a small portion of the database, with the prebuilt index enabling rapid retrieval of the relevant candidate subset.

**Complexity.** For a query graph $G$ and a candidate $\tilde{G}$, the per-query time complexity is

$$O\Big(K\left[\log|\mathcal{B}_{\neg y}| + \alpha M \log(\alpha M)\right] \,+\, d\,\alpha M \,+\, MI|V||\tilde{V}|\Big). \tag{4}$$

The first term arises from CDI, which uses KD-tree search over $K$ prototypes to select $\alpha M$ candidates from a bucket of size $|\mathcal{B}_{\neg y}|$. The second term corresponds to CSI, computing pairwise distances in $d$-dimensional embeddings with partial sorting. The final term accounts for the Sinkhorn step in OT, requiring $I$ iterations over $|V|$ and $|\tilde{V}|$ nodes. Theorem 3 shows that enlarging the candidate set with both indices monotonically increases the probability of finding the optimal counterfactual. In experiments, we set $\alpha = 3$ and $M = 10$, retaining 30 candidates from CDI and 10 from CSI, which is sufficient to identify counterfactuals.

## 5 Theoretical Analysis

Unlike edit-based distances that are computationally intractable or embedding-based distances that collapse local variations, our hypergraph-based concept distance explicitly captures counterfactual edits at both node and concept levels. This makes it particularly suitable for counterfactual reasoning. In this section, we show that this distance can be efficiently approximated by two complementary indices with certified guarantees. CDI captures bounded-depth homomorphism statistics and provides reliable lower bounds for safe pruning, while CSI reconstructs these features in a prototype space and yields computable upper bounds for similarity search. Together, they establish a sandwich bound on the true distance, yielding guarantees on query success rate, ranking correctness, and database-scale efficiency (see also Theorems 5–6 in Appendix B, which provide detailed homomorphism-based analysis of why C²GQ can find frequent subgraphs as concepts).

**Lower bounds via CDI.** We first show that differences in CDI yield certified lower bounds on the hypergraph-based concept distance.

**Theorem 1.** *Let* $\lambda = \min\left\{\lambda_{\text{del}}^V, \lambda_{\text{ins}}^V, \frac{\lambda_{\text{sub}}^V}{2}\right\} + \alpha \cdot \min\left\{\lambda_{\text{del}}^F, \lambda_{\text{ins}}^F, \frac{\lambda_{\text{sub}}^F}{2}\right\}$. *Then,*

$$\Delta(G, \tilde{G}) \,\geq\, \lambda\,\|\boldsymbol{\sigma}_G - \boldsymbol{\sigma}_{\tilde{G}}\|_1.$$

*Here* $\alpha = 1/(r\rho_{\max})$, *where* $r$ *is the maximum factor arity (i.e., nodes per factor) and* $\rho_{\max}$ *is the maximum factor-load of any node (i.e., factors per node).*

Theorem 1 formalizes the intuition that every edit necessarily modifies at least one WL-type count. Intuitively, $\alpha$ is a conservative scaling ensuring that factor-level edits are fully reflected by concept changes in the original graph $G$. Hence, the $\ell_1$ gap between CDIs provides a certified lower bound on edit cost, ensuring that pruning by CDI never discards valid counterfactuals.

**Upper bounds via CSI.** We now introduce CSI as a complementary continuous perspective that provides upper bounds on the hypergraph-based distance.

**Theorem 2.** *Assume each factor feature is given by a permutation-invariant, $L_\psi$-Lipschitz aggregator* $\mathbf{g}_f^G = \psi(\{\mathbf{h}_v^G : v \in f\})$, *and let* $r$ *be the maximum factor arity,* $\rho_{\max}$ *the maximum*

*factor-load of any node, and $\mathbf{C} \in \mathbb{R}^{d \times K}$ the prototype matrix with pseudoinverse $\mathbf{C}^{+}$. Then,*

$$\Delta(G, \tilde{G}) \leq \left( \tfrac{\lambda_{\mathrm{sub}}^{V}}{2} |V_G| + \lambda_{\mathrm{sub}}^{F} \rho_{\max} r L_\psi \right) \|\mathbf{C}^{+}\|_{1\leftarrow 2} \|\mathbf{z}_G - \mathbf{z}_{\tilde{G}}\|_2$$

$$+ \max\{\lambda_{\mathrm{del}}^{V}, \lambda_{\mathrm{ins}}^{V}\} \left| |V_G| - |V_{\tilde{G}}| \right| + \max\{\lambda_{\mathrm{del}}^{F}, \lambda_{\mathrm{ins}}^{F}\} \left| |F_G| - |F_{\tilde{G}}| \right|.$$

$\|\mathbf{C}^{+}\|_{1\leftarrow 2} := \sup_{x \neq 0} \| \mathbf{C}^{+} x \|_1 / \|x\|_2$ *converts the CSI's $\ell_2$ gap into an $\ell_1$ mass movement.*

Theorem 2 shows that CSI enables counterfactual analysis by quantifying smooth trajectories between a query graph and its perturbations, while also supporting concept mining by clustering structurally similar subgraphs. The CSI gap provides a computable upper bound: differences in CSI encode the amount of "semantic mass" to be shifted, certifying feasibility without solving the optimal transport problem exactly.

**Remark.** By combining the Theorems 1 and 2, we obtain a sandwich bound (Corollary 1 in Appendix B.4) that certifies counterfactual queries with three guarantees: (i) query success rates improve monotonically, since enlarging the CSI candidate set cannot reduce recall (Theorem 3 in Appendix B.5); (ii) pairwise rankings are provably correct, as CDI and CSI jointly certify margins between candidates (Theorem 4 in Appendix B.6); and (iii) pruning is safe, since no valid counterfactual lies outside the retained set (Corollary 2 in Appendix B.7).

## 6 Experiments

We evaluate $\mathrm{C}^2\mathrm{GQ}$ on eight datasets spanning molecular, social, biological, and traffic domains. Experimental results demonstrate that $\mathrm{C}^2\mathrm{GQ}$ consistently outperforms strong baselines: it improves query accuracy by over $20\%$, achieves up to a $20\times$ speed-up, and identifies counterfactuals that are both interpretable and domain-valid. Additional details on the experimental setup, dataset statistics, ablation studies, scalability analysis, and sensitivity tests are provided in Appendix C.

### 6.1 Setup

**Datasets.** We evaluate on eight datasets spanning four domains. In the molecular domain, we use *NCI1* (Wale et al., 2008), *Mutag* (Maron & Ames, 1983), and *AIDS* (Ivanov et al., 2019) for binary classification of chemical properties such as anticancer activity, mutagenicity, and HIV activity. In the social domain, we use *Reddit-Binary* (Yanardag & Vishwanathan, 2015), which consists of graphs of online discussion threads. In the biological domain, we evaluate on *PROTEINS* and *ENZYMES* (Borgwardt et al., 2005), covering enzyme classification and protein function prediction. In the traffic domain, we construct subgraph datasets from *METR-LA* and *PEMS-BAY* (Li et al., 2018) for predicting local congestion patterns. Since no benchmark provides ground-truth counterfactuals (Ying et al., 2019; Lucic et al., 2022; Zhang et al., 2023), we follow Giorgi et al. (2025) to synthesize them. Perturbations are designed as minimal yet valid edits to edges or node features that flip predictions while respecting domain constraints. We frame evaluation as a query task: given a graph $G$, the system must rank and retrieve its counterfactual $\tilde{G}^*$.

**Evaluation Protocol.** We compare $\mathrm{C}^2\mathrm{GQ}$ against several graph query baselines. *GED* applies graph edit distance with bipartite and anchor-aware bounds (Riesen & Bunke, 2009; Chang et al., 2017), producing accurate alignments but at high computational cost. Embedding-based methods include *GCN* (Kipf & Welling, 2017), *GIN* (Xu et al., 2019), *DiffPool* (Ying et al., 2018), *SAGPool* (Lee et al., 2019), *Graphormer* (Ying et al., 2021), and *SimGNN* (Bai et al., 2019). For scalability, we also evaluate a *+LSH* variant of each method, which accelerates search by $5$–$10\times$ with only modest accuracy loss. We assess query quality using *Recall@k* and *Mean Reciprocal Rank (MRR)* (Wu et al., 2022), which capture coverage (whether a ground-truth counterfactual appears), position (average rank), and consistency (ranking quality). To evaluate the correctness of predicted concept-edit sets, we report their alignment with ground-truth transformations as *counterfactuals accuracy*, measured by *Precision (P)*, *Recall (R)*, and *F1-score (F1)* (Giorgi et al., 2025). Efficiency is measured by query latency $T$, defined as the average runtime (in seconds) per 100 queries.

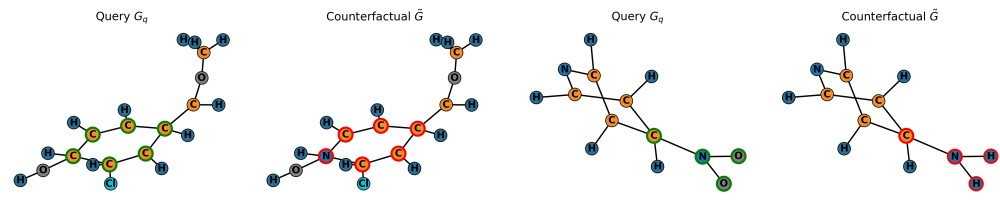

(a) benzene ring → pyridine group      (b) nitro group → amine group

Figure 1: Examples of counterfactual graph queries on the Mutag dataset.

| Method | Molecular | | | | | | | | | Social | | |
| | NCI1 | | | Mutag | | | AIDS | | | Reddit-Binary | | |
| | R@1 | MRR | T | R@1 | MRR | T | R@1 | MRR | T | R@1 | MRR | T |
|---|---|---|---|---|---|---|---|---|---|---|---|---|
| GED | 0.460 | 0.579 | 410 | 0.449 | 0.566 | 22 | 0.473 | 0.590 | 36 | 0.424 | 0.561 | 1020 |
| GCN | 0.504 | 0.612 | 75 | 0.489 | 0.600 | 4 | 0.516 | 0.625 | 8 | 0.465 | 0.591 | 65 |
| GIN | 0.542 | 0.649 | 68 | 0.527 | 0.635 | 4 | 0.558 | 0.658 | 8 | 0.488 | 0.604 | 58 |
| DiffPool | 0.572 | 0.653 | 80 | 0.553 | 0.648 | 3 | 0.590 | 0.673 | 10 | 0.510 | 0.624 | 50 |
| SAGPool | 0.565 | 0.648 | 70 | 0.560 | 0.654 | 2 | 0.582 | 0.670 | 11 | 0.517 | 0.629 | 46 |
| Graphormer | 0.556 | 0.651 | 74 | 0.540 | 0.641 | 3 | 0.573 | 0.664 | 9 | 0.500 | 0.613 | 55 |
| SimGNN | 0.584 | 0.662 | 72 | 0.568 | 0.666 | 2 | 0.601 | 0.688 | 9 | 0.524 | 0.635 | 42 |
| SimGNN+LSH | 0.557 | 0.659 | **15** | 0.544 | 0.647 | **1** | 0.574 | 0.668 | **2** | 0.497 | 0.611 | **13** |
| C²GQ | 0.704 | 0.790 | 29 | 0.683 | 0.773 | 3 | 0.724 | 0.808 | 5 | 0.628 | 0.728 | 42 |
| C²GQ (w/o index) | **0.710** | **0.795** | 520 | **0.690** | **0.780** | 30 | **0.730** | **0.813** | 55 | **0.634** | **0.734** | 1323 |
| Method | Biological | | | | | | Traffic | | | | | |
| | PROTEINS | | | ENZYMES | | | METR-LA | | | PEMS-BAY | | |
| | R@1 | MRR | T | R@1 | MRR | T | R@1 | MRR | T | R@1 | MRR | T |
| GED | 0.440 | 0.576 | 138 | 0.409 | 0.547 | 95 | 0.405 | 0.540 | 470 | 0.413 | 0.555 | 722 |
| GCN | 0.475 | 0.600 | 24 | 0.445 | 0.573 | 16 | 0.427 | 0.561 | 60 | 0.435 | 0.569 | 85 |
| GIN | 0.503 | 0.616 | 21 | 0.470 | 0.590 | 13 | 0.446 | 0.581 | 54 | 0.454 | 0.588 | 78 |
| DiffPool | 0.520 | 0.627 | 22 | 0.480 | 0.592 | 14 | 0.452 | 0.584 | 47 | 0.468 | 0.597 | 70 |
| SAGPool | 0.512 | 0.622 | 20 | 0.488 | 0.598 | 11 | 0.460 | 0.589 | 45 | 0.462 | 0.594 | 67 |
| Graphormer | 0.511 | 0.620 | 23 | 0.478 | 0.589 | 12 | 0.452 | 0.583 | 55 | 0.460 | 0.592 | 74 |
| SimGNN | 0.529 | 0.632 | 19 | 0.493 | 0.605 | 12 | 0.463 | 0.596 | 44 | 0.471 | 0.602 | 64 |
| SimGNN+LSH | 0.500 | 0.610 | **6** | 0.470 | 0.585 | **4** | 0.438 | 0.573 | **15** | 0.445 | 0.580 | **20** |
| C²GQ | 0.632 | 0.763 | 11 | 0.623 | 0.748 | 7 | 0.621 | 0.716 | 34 | 0.635 | 0.724 | 48 |
| C²GQ (w/o index) | **0.648** | **0.779** | 200 | **0.640** | **0.765** | 140 | **0.640** | **0.733** | 652 | **0.672** | **0.758** | 927 |

Table 1: Query-level retrieval performance grouped by domain.

**Implementation Details.** Concept embeddings are obtained from a pretrained GCN (Lu et al., 2021) with hidden dimensionality 128, depth $L = 2$, and batch size 64, trained using Adam with a learning rate of $10^{-3}$. Final-layer node embeddings $\{\mathbf{h}_v\}$ are clustered into $K = 64$ prototypes via $K$-means, yielding for each graph in $\mathcal{D}$ its CDI $\boldsymbol{\sigma}_G$ and CSI $\mathbf{z}_G$. Penalty weights $\lambda$ in the hypergraph-based distance are estimated from dataset statistics to balance edits across concepts. Retrieval is performed in three stages: (i) **Label hashing** to discard graphs with mismatched labels, (ii) **Dual indexing** on $\boldsymbol{\sigma}_G$ and $\mathbf{z}_G$ for coarse pruning ($\alpha = 3$, $M = 10$), and (iii) **Fine re-ranking** of remaining candidates via the hypergraph-based concept distance with entropic Sinkhorn iterations ($\varepsilon = 0.01$, max 200).

## 6.2 EXPERIMENTAL RESULTS

**Case Study.** Unlike prior methods that reduce counterfactuals to isolated node- or edge-level perturbations, C²GQ identifies coherent, concept-level counterfactuals grounded in the database. As illustrated in Fig. 1, the highlighted regions (green in the query graph and red in the counterfactual) correspond to chemically meaningful substructures rather than arbitrary atom-level edits—for example, substituting a benzene ring with a pyridine ring or replacing a nitro group with an amine group, both of which induce plausible functional modifications (Reiser et al., 2022). These findings demonstrate that C²GQ uncovers interpretable, domain-valid counterfactual trajectories at the functional-group level, yielding explanations that are not only chemically meaningful but also informative of model behavior.

**Validation Study.** To evaluate the contribution of learned concepts, we conducted a control experiment by randomly permuting candidate graph concepts prior to retrieval. This procedure preserves the size of the concept space while destroying semantic alignment between prototypes and substructures. Relative to this randomized baseline, C²GQ achieved

| Method | Molecular | | | | | | | | | Social | | |
|---|---|---|---|---|---|---|---|---|---|---|---|---|
| | NCI1 | | | Mutag | | | AIDS | | | Reddit-Binary | | |
| | R | P | F1 | R | P | F1 | R | P | F1 | R | P | F1 |
| GED | 0.552 | 0.534 | 0.543 | 0.566 | 0.548 | 0.557 | 0.574 | 0.556 | 0.565 | 0.493 | 0.478 | 0.485 |
| SimGNN | 0.711 | 0.694 | 0.702 | 0.726 | 0.707 | 0.716 | 0.739 | 0.720 | 0.729 | 0.591 | 0.574 | 0.582 |
| $C^2$GQ | **0.805** | **0.839** | **0.822** | 0.807 | **0.781** | 0.794 | **0.835** | 0.806 | **0.820** | 0.671 | 0.656 | 0.663 |
| $C^2$GQ (w/o index) | 0.798 | 0.834 | 0.816 | **0.858** | 0.781 | **0.819** | 0.827 | **0.806** | 0.816 | **0.678** | **0.660** | **0.669** |

| Method | Biological | | | | | | Traffic | | | | | |
|---|---|---|---|---|---|---|---|---|---|---|---|---|
| | PROTEINS | | | ENZYMES | | | METR-LA | | | PEMS-BAY | | |
| | R | P | F1 | R | P | F1 | R | P | F1 | R | P | F1 |
| GED | 0.491 | 0.472 | 0.481 | 0.458 | 0.442 | 0.450 | 0.426 | 0.411 | 0.418 | 0.414 | 0.398 | 0.406 |
| SimGNN | 0.609 | 0.591 | 0.600 | 0.573 | 0.555 | 0.564 | 0.532 | 0.514 | 0.523 | 0.519 | 0.502 | 0.510 |
| $C^2$GQ | **0.697** | 0.672 | **0.684** | 0.655 | **0.637** | 0.646 | **0.619** | 0.601 | **0.610** | 0.598 | 0.589 | 0.593 |
| $C^2$GQ (w/o index) | 0.693 | **0.676** | 0.684 | **0.708** | 0.629 | **0.667** | 0.614 | **0.601** | 0.608 | **0.656** | 0.593 | **0.623** |

Table 2: Counterfactual set accuracy grouped by domain.

an average improvement of 27% in F1 across datasets. A paired $t$-test confirmed the statistical significance of this gain ($p \approx 0.029$), indicating that the improvements arise from structured concept modeling rather than chance. These results demonstrate that $C^2$GQ uncovers domain-valid mechanisms, underscoring its value in high-stakes applications such as drug discovery and financial risk

**Query Accuracy and Efficiency.** Table 1 shows that $C^2$GQ consistently outperforms the strongest baseline (SimGNN), with average gains of 12.4% in MRR and 13.5% in Recall@1. These improvements highlight the benefit of concept-level modeling in capturing counterfactual similarity, yielding robust generalization across molecular, social, biological, and traffic domains. The advantage is most pronounced on traffic datasets, where $C^2$GQ achieves more than 20% higher accuracy. In contrast, GED computes edit distance directly on nodes and edges, requiring full OT alignment without concept abstraction, which makes it both slow and semantically limited. On Reddit-Binary, for example, GED takes over 1000 seconds yet still underperforms $C^2$GQ in ranking. Embedding-based methods are faster than GED but, lacking concept-awareness, lag by more than 15% in accuracy. LSH-based variants further accelerate queries but incur substantial accuracy loss, underscoring the efficiency–accuracy trade-off. By contrast, our two concept-aware indices approximate the hypergraph-based distance with certified bounds (Theorems 3 and 4), achieving an average $20\times$ speedup with less than a 1% accuracy drop. This demonstrates that indexing is not only practically effective but also theoretically grounded. Finally, scalability tests (Appendix C.3) show that $C^2$GQ grows sublinearly with graph size $|G|$ and dataset size $|\mathcal{D}|$, whereas GED scales quadratically and linearly, respectively.

**Counterfactuals Accuracy.** Table 2 reports *counterfactuals accuracy*, which measures the alignment between predicted edits and ground-truth transformations. Overall, $C^2$GQ achieves the highest F1, improving by an average of 44% over GED and 14% over SimGNN. These gains arise from jointly learning shared embeddings and constructing concept-prototypical sets, which provide a stable foundation for counterfactual reasoning. By leveraging the hypergraph-based concept distance, our method captures both fine-grained and coarse-grained structural variations. Moreover, the indices are theoretically guaranteed to preserve essential subgraph information via graph homomorphisms (Theorems 5 and 6), enabling scalability without sacrificing fidelity and interpretability.

# 7 Conclusions

We introduced the **Counterfactual Graph Database (CF-GDB)**, the first retrieval-based framework that grounds counterfactual reasoning in verifiable graph instances. At its core, **Concept-Based Counterfactual Graph Query ($C^2$GQ)** abstracts graphs into semantic concepts and measures differences via a hypergraph-based distance, accelerated by two indices (CDI and CSI) with certified bounds. Our theoretical analysis established sandwich guarantees, while experiments across four domains demonstrated over 20% accuracy gains and up to $20\times$ speedups against strong baselines. CF-GDB thus provides a principled foundation for scalable, interpretable, and domain-valid counterfactual analysis, paving the way for conditional queries, fairness auditing, and other high-stakes applications.

## Reproducibility Statement

Implementation details such as concept extraction, hypergraph-based distance, indexing strategies (CDI/CSI), and retrieval pipeline pseudocode are presented in Appendix A. All theoretical contributions, including formal definitions of counterfactual queries, certified bounds, and proofs of correctness and efficiency (Theorems 1-6), are provided in detail in Appendix B to enable independent verification. For empirical evaluation, we describe datasets across four domains (molecular, social, biological, and traffic) and the associated tasks in Section 6, with additional dataset statistics, hyperparameters, and ablation studies reported in Appendix C. To further enhance transparency, we release our implementation as anonymous supplementary material at `https://anonymous.4open.science/r/CF-GDB-2DD6`, which includes scripts for preprocessing datasets, computing indices, and executing $C^2GQ$ queries. For theoretical claims, all assumptions are explicitly stated, and complete proofs are provided for guarantees on query success, ranking correctness, and safe pruning.

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

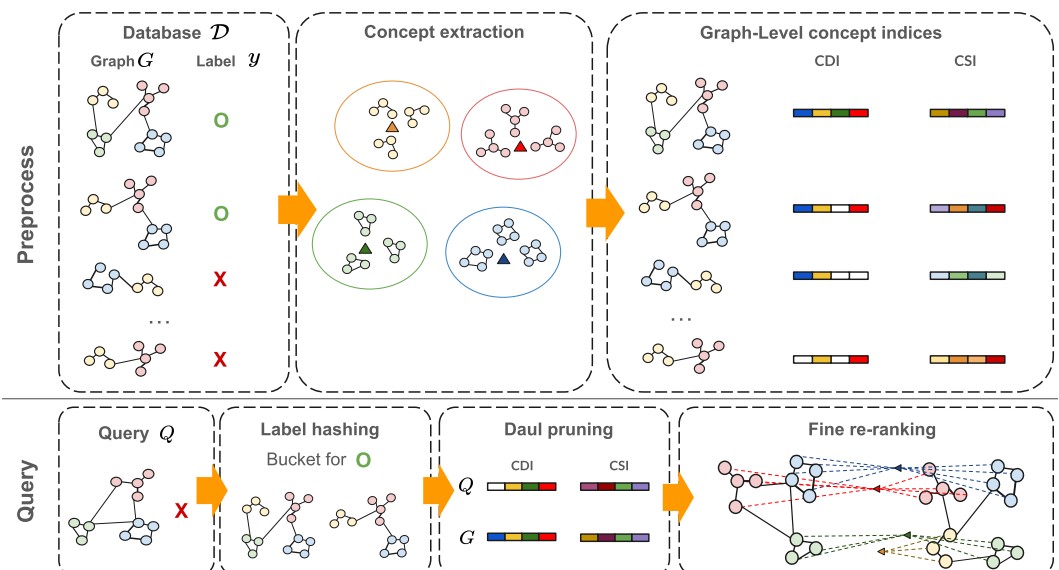

Figure 2: Overview of the C²GQ framework.

## A  Detail of Concept-based Counterfactual Graph Query

We begin by formally defining the *concept assignment*, a mapping learned across multiple graphs in the database that projects nodes into a shared vocabulary of semantic proto-types. This induces concept groups globally, in contrast to prior methods that operate on single graphs (Ying et al., 2019; Verma et al., 2024a; Abrate & Bonchi, 2021; Lucic et al., 2022). Building on these assignments, we introduce a *hypergraph-based concept distance* that jointly aligns nodes and concept-level factors, capturing edits at multiple granularities while respecting domain-specific constraints (e.g., chemical valency or structural consistency). To ensure scalability, we further propose two lightweight *concept-based indices*—the Concept Distribution Index (CDI) and the Concept Semantic Index (CSI)—which approximate the hypergraph distance with certified lower and upper bounds. Together, these indices enable efficient yet faithful counterfactual search. The notations are summarized in Table 3.

As illustrated in Fig. 2, our C²GQ framework operates in two stages. The first is prepro-cessing, where we learn and extract graph concepts by applying K-means clustering ($K = 4$, shown as differently colored triangles) to group structurally similar subgraphs across multi-ple graphs in the database $\mathcal{D}$. Each cluster centroid, represented by a triangle, serves as a prototype concept. These concepts are then used to construct two indices per query, which are stored in a KD-tree for efficient retrieval.

In the query stage, we proceed in two steps. First, during label hashing, we filter the counterfactual bucket—i.e., graphs whose labels differ from that of the query graph. Second, we extract the indices of the query graph and compare them against candidates in the counterfactual bucket. Finally, we compute the hypergraph-based concept distance to rerank the candidates and identify the optimal counterfactual.

### A.1  Concept Extraction via GNNs.

To effectively extract concepts, we employ Graph Neural Networks (GNNs) (Kipf & Welling, 2017; Veličković et al., 2018) to compute structure-aware node embeddings. At layer $\ell$, each node $v \in V_G$ updates its representation through a message-passing scheme:

$$\mathbf{m}_v^{(\ell)} = \text{AGGREGATE}^{(\ell)}\Big(\big\{\,\mathbf{h}_u^{(\ell-1)} : u \in N(v)\,\big\}\Big), \quad \mathbf{h}_v^{(\ell)} = \text{COMBINE}^{(\ell)}\Big(\mathbf{h}_v^{(\ell-1)}, \mathbf{m}_v^{(\ell)}\Big), \quad (5)$$

| Symbol | Meaning | Type / Shape |
|---|---|---|
| | *Graph* | |
| $G = (V_G, E_G, X_G)$ | graph with nodes, edges, node features | $X_G \in \mathbb{R}^{|V_G| \times d}$ |
| $y \in \mathcal{Y}$ | class label of $G$ | finite label set |
| $\mathcal{D} = \{(G, y)\}$ | graph database (graphs with labels) | set |
| $Q = (G, y)$ | query (graph, label) | pair |
| $\tilde{G}$ | candidate graph (counterfactual) | graph |
| | *Hypergraph* | |
| $H_G = (V_G, F_G)$ | hypergraph over concepts (factors) | nodes $V_G$, factors $F_G$ |
| $f \in F_G$ | factor (group of nodes sharing $a(\cdot)$) | set of nodes |
| $e(f)$ | incident node set of factor $f$ | subset of $V_G$ |
| $\boldsymbol{g}_f = \psi(\{h_v : v \in f\})$ | factor embedding via aggregator $\psi$ | $\mathbb{R}^d$ |
| $\psi$ | permutation-invariant aggregator (mean/sum/attn) | $\mathbb{R}^{d \times |e(f)|} \to \mathbb{R}^d$ |
| $L_\psi$ | Lipschitz constant of $\psi$ | scalar |
| $r$ | max factor arity ($|e(f)| \leq r$) | integer |
| $\rho(v)$ | factor-load of node $v$ | integer |
| $\rho_{\max}$ | $\max_v \rho(v)$ | integer |
| | *Concepts embedding* | |
| $h_v$ | node embedding of $v$ | $\mathbb{R}^d$ |
| $d$ | embedding dimension | integer |
| $H = \{h_v\}$ | set of node embeddings over $\mathcal{D}$ | multiset in $\mathbb{R}^d$ |
| $K$ | # concept prototypes (vocabulary size) | integer |
| $\{c_k\}_{k=1}^K$ | concept prototypes (cluster centroids) | $c_k \in \mathbb{R}^d$ |
| $C = [c_1, \ldots, c_K]$ | prototype matrix | $\mathbb{R}^{d \times K}$ |
| $C^+$ | pseudoinverse of $C$ | $\mathbb{R}^{K \times d}$ |
| $a(v) \in [K]$ | concept assignment of node $v$ | index |
| | *Hypergraph-based concept distance* | |
| $\Delta(G, \tilde{G})$ | hypergraph-based concept distance | scalar $\geq 0$ |
| $T = \begin{bmatrix} T_{V_G V_{\tilde{G}}} & 0 \\ 0 & T_{F_G F_{\tilde{G}}} \end{bmatrix}$ | transport plan (nodes & factors) | nonnegative matrices |
| $\delta_V, \; \delta_F$ | node/factor matching costs (e.g., $\ell_2^2$) | $\mathbb{R}_{\geq 0}$ |
| $\lambda_V^{\mathrm{sub}}, \lambda_V^{\mathrm{del}}, \lambda_V^{\mathrm{ins}}$ | node substitution/deletion/insertion penalties | $\mathbb{R}_{\geq 0}$ |
| $\lambda_F^{\mathrm{sub}}, \lambda_F^{\mathrm{del}}, \lambda_F^{\mathrm{ins}}$ | factor substitution/deletion/insertion penalties | $\mathbb{R}_{\geq 0}$ |
| $I$ | # Sinkhorn iterations (OT) | integer |
| $\varepsilon$ | entropic regularization (OT) | scalar |
| | *Concept indices* | |
| $\sigma_G(k) = \sum_{v \in V_G} \mathbf{1}[a(v) = k]$ | Concept Distribution Index (CDI) | $\mathbb{N}^K$ |
| $z_G = \sum_{k=1}^K \frac{\sigma_G(k)}{|V_G|} c_k$ | Concept Semantic Index (CSI) | $\mathbb{R}^d$ |
| $\delta_{\mathrm{freq}}(G, \tilde{G}) = \|\sigma_G - \sigma_{\tilde{G}}\|_1$ | histogram (CDI) distance | scalar |
| $\delta_{\mathrm{sem}}(G, \tilde{G}) = \|z_G - z_{\tilde{G}}\|_2$ | semantic (CSI) distance | scalar |
| $L(G, \tilde{G})$ | CDI-based certified lower bound | scalar |
| $U(G, \tilde{G})$ | CSI-based certified upper bound | scalar |
| $\|C^+\|_{1 \leftarrow 2} = \sup_{x \neq 0} \frac{\|C^+ x\|_1}{\|x\|_2}$ | operator norm for CSI bound | scalar |
| $\alpha = \frac{1}{r \, \rho_{\max}}$ | scaling constant in CDI lower bound | scalar |
| | *Querying* | |
| $\{\mathcal{B}_y\}_{y \in \mathcal{Y}}$ | label-specific buckets | partition of $\mathcal{D}$ |
| $\mathcal{B}_{\neg y} = \bigcup_{y' \in \mathcal{Y}, \, y' \neq y} \mathcal{B}_{y'}$ | union of buckets with labels different from $y$ | subset of $\mathcal{D}$ |
| $N_{\neg y} = |\mathcal{B}_{\neg y}|$ | size of $\mathcal{B}_{\neg y}$ | integer |
| $M$ | # retained candidates after pruning | integer |
| $\alpha$ (prune) | pruning ratio in dual indexing | scalar |

Table 3: Notation Table.

where AGGREGATE summarizes neighbor information and COMBINE integrates it with the node's own state. After $L$ layers, we obtain the final node embeddings $\mathbf{h}_v^{(L)}$ (or simply $\mathbf{h}_v$), and denote the collection across the database as

$$\mathcal{H} = \{\mathbf{h}_v \mid v \in V_G, \; G \in \mathcal{D}\}. \tag{6}$$

We then cluster $\mathcal{H}$ into $K$ prototypes via $K$-means, obtaining a global semantic dictionary $\mathbf{C} = [\mathbf{c}_1, \ldots, \mathbf{c}_K]$. This dictionary amortizes computation by shifting clustering offline and provides a shared semantic basis for efficient and meaningful distance computation in counterfactual graph queries.

**Definition 1** (Concept Assignment)**.** *Given prototypes* $\mathbf{C}$, *a concept assignment for a graph* $G = (V_G, E_G)$ *is a mapping*

$$a : V_G \rightarrow \{1, \ldots, K\}, \quad a(v) = \arg \min_{k \in [K]} \|\mathbf{h}_v - \mathbf{c}_k\|_2^2, \tag{7}$$

*which assigns each node* $v \in V_G$ *to its closest prototype.*

The assignment $a$ induces a partition $\{V_G^k\}_{k=1}^K$, grouping nodes that share the same concept. These groups form the building blocks for higher-level structures such as concept histograms (CDI), semantic embeddings (CSI), and concept-level hyperedges in the hypergraph representation.

## A.2 HYPERGRAPH-BASED CONCEPT DISTANCE

Having defined concept assignments, we now show how they induce a hypergraph representation of a graph, which in turn enables a distance measure that jointly aligns node-level and concept-level structures.

Each graph $G$ (and $\tilde{G}$) is mapped to a hypergraph $H_G = (V_G, F_G)$, where each factor $f \in F_G$ groups nodes sharing the same concept assignment $a(v)$. Factor embeddings are computed as

$$\mathbf{g}_f^G = \psi\big(\{\mathbf{h}_v^G : v \in f\}\big),$$

where $\psi$ is a permutation-invariant operator such as mean, sum, or attention (Xu et al., 2019; Lee et al., 2019). In this way, factor representations capture higher-order dependencies—such as communities in social networks or functional groups in molecules—that cannot be represented by pairwise edges.

We define a block-diagonal transport plan

$$T = \begin{bmatrix} T^{V_G V_{\tilde{G}}} & 0 \\ 0 & T^{F_G F_{\tilde{G}}} \end{bmatrix}, \qquad T^{V_G V_{\tilde{G}}}, T^{F_G F_{\tilde{G}}} \geq 0,$$

where $T_{v,\tilde{v}}^{V_G V_{\tilde{G}}}$ denotes the transported mass from node $v \in V_G$ to $\tilde{v} \in V_{\tilde{G}}$, and analogously for factors. Standard partial matching constraints (Chapel et al., 2020) enforce valid alignments, disentangling node- and concept-level distances and yielding interpretable edits such as removals or substitutions.

**Definition 2** (Hypergraph-based Concept Distance)**.** *Given two graphs* $G$ *and* $\tilde{G}$, *the hypergraph-based concept distance is defined as*

$$\Delta(G, \tilde{G}) = \min_{T^{V_G V_{\tilde{G}}}, T^{F_G F_{\tilde{G}}} \geq 0} \; \lambda_{\text{sub}}^V \sum_{v,v'} T_{vv'}^{V_G V_{\tilde{G}}} \delta_V(\mathbf{h}_v^G, \mathbf{h}_{v'}^{\tilde{G}}) + \lambda_{\text{del}}^V\Big(|V_G| - \sum_{v,v'} T_{vv'}^{V_G V_{\tilde{G}}}\Big) + \lambda_{\text{ins}}^V\Big(|V_{\tilde{G}}| - \sum_{v,v'} T_{vv'}^{V_G V_{\tilde{G}}}\Big)$$
$$+ \lambda_{\text{sub}}^F \sum_{f,f'} T_{ff'}^{F_G F_{\tilde{G}}} \delta_F(\mathbf{g}_f^G, \mathbf{g}_{f'}^{\tilde{G}}) + \lambda_{\text{del}}^F\Big(|F_G| - \sum_{f,f'} T_{ff'}^{F_G F_{\tilde{G}}}\Big) + \lambda_{\text{ins}}^F\Big(|F_{\tilde{G}}| - \sum_{f,f'} T_{ff'}^{F_G F_{\tilde{G}}}\Big). \tag{8}$$

*subject to the partial matching constraints*

$$\sum_{v'} T_{vv'}^{V_G V_{\tilde{G}}} \leq 1, \quad \sum_v T_{vv'}^{V_G V_{\tilde{G}}} \leq 1, \quad \sum_{f'} T_{ff'}^{F_G F_{\tilde{G}}} \leq 1, \quad \sum_f T_{ff'}^{F_G F_{\tilde{G}}} \leq 1.$$

Here, $\lambda_{\text{sub}}^V, \lambda_{\text{del}}^V, \lambda_{\text{ins}}^V, \lambda_{\text{sub}}^F, \lambda_{\text{del}}^F, \lambda_{\text{ins}}^F \geq 0$ are penalties for substitution, deletion, and insertion at the node and factor levels, while $\delta_V$ and $\delta_F$ denote squared $\ell_2$ costs.

These penalty weights are estimated from data by: (i) sampling graphs from the database, (ii) applying controlled micro-edits (e.g., node deletion, insertion with sampled attributes, or feature substitution via nearest neighbors), (iii) recording the induced changes in concept histograms, and (iv) fitting a linear or nonnegative least-squares regression that maps histogram shifts to unit edit costs. The fitted coefficients then provide calibrated estimates of each $\lambda$.

## A.3 CONCEPT-BASED INDICES

While the hypergraph-based distance provides a flexible notion of similarity, computing it exactly can be expensive. To enable scalable search, we introduce lightweight indices that approximate this distance while preserving key guarantees.

**Definition 3** (Concept-based Indices). *Given a graph $G$ with concept assignment $a : V_G \to [K]$, we define:*

- **Concept Distribution Index (CDI):** $\sigma_G(k) = \sum_{v \in V_G} \mathbf{1}\{a(v) = k\}, \quad k \in [K]$, *which counts the number of nodes assigned to each concept.*
- **Concept Semantic Index (CSI):** $\mathbf{z}_G = \sum_{k=1}^{K} \frac{\sigma_G(k)}{|V_G|} \mathbf{c}_k$, *where $\mathbf{c}_k \in \mathbb{R}^d$ is the embedding of concept $k$.*

CDI captures discrete structural edits such as insertions, deletions, or substitutions, whereas CSI encodes the overall semantic distribution of concepts and varies smoothly under small perturbations (e.g., minor attribute changes or connectivity edits). Together, CDI and CSI provide complementary coarse representations that enable efficient yet faithful approximation of the hypergraph-based distance.

## A.4 PSEUDOCODE

---

**Algorithm 1** C$^2$GQ Prepossessing (Offline)

---

1: **Input:** Graph database $\mathcal{D} = \{(G, y)\}$; embedding dim $d$; prototypes $K$
2: **Output:** Prototypes $C = [c_1, \ldots, c_K]$; per-graph indices $\{(\sigma_G, z_G)\}$
3: Train node encoder (GNN) on $\mathcal{D}$ to obtain embeddings $\{h_v\}$
4: Collect all node embeddings $H \leftarrow \{h_v\}$
5: Run K-means on $H$ to obtain prototypes $\{c_k\}$ and assignments $a(v)$
6: **for** each graph $G \in \mathcal{D}$ **do**
7: $\quad \sigma_G(k) \leftarrow \sum_{v \in V_G} \mathbf{1}[a(v) = k]$ {CDI (histogram)}
8: $\quad z_G \leftarrow \sum_{k=1}^{K} \frac{\sigma_G(k)}{|V_G|} c_k$ {CSI (semantic embedding)}
9: **end for**
10: Return $C$, $\{(\sigma_G, z_G)\}_{G \in \mathcal{D}}$

---

**Algorithm 2** C$^2$GQ Query (Online)

---

1: **Input:** Query $Q = (G, y)$; prototypes $C$; indices $\{(\sigma_{G'}, z_{G'})\}$; buckets $\{B_{y'}\}$; shortlist size $M$
2: **Output:** Top-$M$ counterfactual candidates
3: $\mathcal{C} \leftarrow \bigcup_{y' \neq y} B_{y'}$ {Opposite-label search space}
4: Compute $(\sigma_G, z_G)$ for query $G$
5: **for** each $G' \in \mathcal{C}$ **do**
6: $\quad \delta_{\text{freq}}(G, G') \leftarrow \|\sigma_G - \sigma_{G'}\|_1$
7: **end for**
8: Keep $\alpha \cdot M$ smallest $\delta_{\text{freq}}$ candidates $\mathcal{C}_1$
9: **for** each $G' \in \mathcal{C}_1$ **do**
10: $\quad \delta_{\text{sem}}(G, G') \leftarrow \|z_G - z_{G'}\|_2$
11: **end for**
12: Keep $M$ smallest $\delta_{\text{sem}}$ candidates $\mathcal{C}_2$
13: Optionally re-rank $\mathcal{C}_2$ with final distance $\Delta(G, G')$
14: Return $\mathcal{C}_2$ sorted by $\Delta(G, G')$

---

# B   Detailed Proof

## B.1   Graph Homomorphism

We then by introducing bounded-depth rooted homomorphisms, a notion fundamental to subgraph (concept) mining (Amini et al., 2012).

**Definition 4** (Rooted Homomorphism). *Let $(R, \text{root})$ be a rooted tree and let $G$ be a graph. A rooted homomorphism from $(R, \text{root})$ to $(G, v)$ is a mapping $\varphi : V_R \to V_G$ such that (i) $\varphi(\text{root}) = v$; (ii) for every $(u, u') \in E_R$, we have $(\varphi(u), \varphi(u')) \in E_G$; and (iii) node labels (if present) are preserved by $\varphi$. The rooted homomorphism count is*

$$\text{hom}_\circ((R, \text{root}), G) := \sum_{v \in V_G} \text{hom}_\circ((R, \text{root}), (G, v)), \tag{9}$$

*i.e., the total number of root-preserving homomorphisms of $R$ into $G$.*

Let $\mathcal{R}_{\leq L}$ denote the set of rooted $L$-types (e.g., rooted $L$-neighborhood computation trees under 1-WL refinement). For a graph $G$, the Concept Distribution Index (CDI) is $\boldsymbol{\sigma}_G \in \mathbb{N}^K$ with $K := |\mathcal{R}_{\leq L}|$, where the $k$-th entry counts the number of nodes of type $c_k$.[1]

Building on this perspective, we establish a linear correspondence between CDI and rooted homomorphism counts (Theorem 5). Having demonstrated the effectiveness of the discrete CDI representation, we next introduce the Concept Semantic Index (CSI) as a complementary continuous view. In particular, we show that CSI recovers bounded-depth homomorphism features, ensuring that the continuous representation retains the expressive power of the discrete statistics (Theorem 6).

## B.2   Proof of Theorem 1

**Theorem 1.** *Let $\lambda = \min\left\{\lambda^V_{\text{del}}, \lambda^V_{\text{ins}}, \frac{\lambda^V_{\text{sub}}}{2}\right\} + \alpha \cdot \min\left\{\lambda^F_{\text{del}}, \lambda^F_{\text{ins}}, \frac{\lambda^F_{\text{sub}}}{2}\right\}$. Then,*

$$\Delta(G, \tilde{G}) \geq \lambda \|\boldsymbol{\sigma}_G - \boldsymbol{\sigma}_{\tilde{G}}\|_1.$$

*Here $\alpha = 1/(r\rho_{\max})$, where $r$ is the maximum factor arity (i.e., nodes per factor) and $\rho_{\max}$ is the maximum factor-load of any node (i.e., factors per node).*

**Assumption 1** (Factor construction and loads). *Each factor $f \in F$ is determined, up to its "type", by a permutation-invariant rule on the multiset of incident node concepts: there exists $\Psi$ such that $\text{type}(f) = \Psi(\{a(u) : u \in f\})$. The factor feature is $\mathbf{g}_f = \psi(\{h_u : u \in f\})$, where $\psi$ is permutation-invariant and $L_\psi$-Lipschitz. Let the factor arity be bounded by $|f| \leq r$ and the node factor-load $\rho(v) := |\{f \in F : v \in f\}|$ be bounded with $\rho_{\max} := \max_v \rho(v) < \infty$.*

**Assumption 2** (Node-local feasibility). *Feasible counterfactual edits are node-local: any change to a factor must be induced by edits to its incident nodes (or their weights). Equivalently, the feasible region forbids "pure factor-only" edits.*

**Assumption 3** (Multiset sensitivity and nondegeneracy). *The map $\Psi$ is strictly sensitive to changes in the incident multiset of node concepts: if $\{a(u) : u \in f\}$ changes, then $\text{type}(f)$ changes. Let $\rho_{\min} := \min_v \rho(v)$, and assume the nondegeneracy condition*

$$r \, \rho_{\max} \, \rho_{\min} \geq 2. \tag{10}$$

For a graph $G$, write its CDI as $\boldsymbol{\sigma}_G \in \mathbb{N}^K$ and define

$$D := \|\boldsymbol{\sigma}_G - \boldsymbol{\sigma}_{\tilde{G}}\|_1. \tag{11}$$

**Lemma 1** (Node edits versus CDI $\ell_1$ gap). *Let $N$ be the minimum number of node-level edits (deletions, insertions, substitutions) to transform $G$ into $\tilde{G}$. Then*

$$N \geq \frac{D}{2}. \tag{12}$$

---

[1] Here $K$ is the dimensionality of the representation. In practice one may also set the number of clusters in $K$-means to $K$, reflecting a partition of nodes into $K$ concept categories.

*Proof.* A deletion (resp. insertion) changes one concept count by $-1$ (resp. $+1$), increasing $D$ by 1. A substitution decreases one count by 1 and increases another by 1, increasing $D$ by 2. Hence each node edit increases $D$ by at least 1 (substitutions by 2), so $2N \geq D$. $\square$

**Lemma 2** (Factor-to-node charging). *Under Assumptions 1–3, the total number of factor changes between $G$ and $\tilde{G}$ satisfies*

$$\#(\text{factor changes}) \ \geq \ \frac{\rho_{\min}}{2}\, D. \tag{13}$$

*Proof.* By Assumption 2, every factor change must be induced by edits on its incident nodes. By Assumption 3, whenever a node's concept changes, *each* incident factor must change its type. Thus one node-concept change induces at least $\rho_{\min}$ factor changes. Let $N$ be the minimal number of node edits; then factor changes $\geq \rho_{\min} N$. By Lemma 1, $N \geq D/2$, so the claim follows. $\square$

*Full proof of the lower bound.* Let

$$\kappa_V := \min\Big\{\lambda_{\mathrm{del}}^V,\, \lambda_{\mathrm{ins}}^V,\, \tfrac{\lambda_{\mathrm{sub}}^V}{2}\Big\}, \qquad \kappa_F := \min\Big\{\lambda_{\mathrm{del}}^F,\, \lambda_{\mathrm{ins}}^F,\, \tfrac{\lambda_{\mathrm{sub}}^F}{2}\Big\}. \tag{14}$$

**Node level.** Each unit increase of the CDI $\ell_1$ gap incurs at least $\kappa_V$ cost from node edits, hence

$$\text{node-cost} \ \geq \ \kappa_V\, D. \tag{15}$$

**Factor level.** Each factor change costs at least $\kappa_F$. By Lemma 2, factor changes $\geq (\rho_{\min}/2)\, D$, hence

$$\text{factor-cost} \ \geq \ \kappa_F \cdot \frac{\rho_{\min}}{2}\, D. \tag{16}$$

Summing (15)–(16) yields

$$\Delta(G, \tilde{G}) \ \geq \ \Big(\kappa_V + \frac{\rho_{\min}}{2}\, \kappa_F\Big)\, D. \tag{17}$$

Now define $\alpha := 1/(r\, \rho_{\max})$. By (10), we have $\alpha \leq \rho_{\min}/2$, hence

$$\kappa_V + \alpha\, \kappa_F \ \leq \ \kappa_V + \frac{\rho_{\min}}{2}\, \kappa_F. \tag{18}$$

Combining with (17) gives

$$\Delta(G, \tilde{G}) \ \geq \ \Big(\kappa_V + \alpha\, \kappa_F\Big) D \ = \ \Big(\min\{\lambda_{\mathrm{del}}^V, \lambda_{\mathrm{ins}}^V, \tfrac{\lambda_{\mathrm{sub}}^V}{2}\} + \alpha\, \min\{\lambda_{\mathrm{del}}^F, \lambda_{\mathrm{ins}}^F, \tfrac{\lambda_{\mathrm{sub}}^F}{2}\}\Big) \big\|\boldsymbol{\sigma}_G - \boldsymbol{\sigma}_{\tilde{G}}\big\|_1. \tag{19}$$

The scaling $\alpha = 1/(r\rho_{\max})$ conservatively converts factor-level edits into CDI $\ell_1$ mass while avoiding over-counting across incident factors. Since at least one penalty is strictly positive, the coefficient is $\lambda > 0$, the theorem follows. $\square$

### B.3  Proof of Theorem 2

**Theorem 2.** *Assume each factor feature is given by a permutation-invariant, $L_\psi$-Lipschitz aggregator $\mathbf{g}_f^G = \psi(\{\mathbf{h}_v^G : v \in f\})$, and let $r$ be the maximum factor arity, $\rho_{\max}$ the maximum factor-load of any node, and $\mathbf{C} \in \mathbb{R}^{d \times K}$ the prototype matrix with pseudoinverse $\mathbf{C}^+$. Then,*

$$\Delta(G, \tilde{G}) \ \leq \ \Big(\tfrac{\lambda_{\mathrm{sub}}^V}{2}\, |V_G| \ + \ \lambda_{\mathrm{sub}}^F\, \rho_{\max}\, r\, L_\psi\Big) \|\mathbf{C}^+\|_{1 \leftarrow 2}\, \|\mathbf{z}_G - \mathbf{z}_{\tilde{G}}\|_2$$

$$+ \ \max\{\lambda_{\mathrm{del}}^V, \lambda_{\mathrm{ins}}^V\}\, \big||V_G| - |V_{\tilde{G}}|\big| \ + \ \max\{\lambda_{\mathrm{del}}^F, \lambda_{\mathrm{ins}}^F\}\, \big||F_G| - |F_{\tilde{G}}|\big|.$$

$\|\mathbf{C}^+\|_{1 \leftarrow 2} := \sup_{x \neq 0} \| \mathbf{C}^+ x \|_1 / \|x\|_2$ *converts the CSI's $\ell_2$ gap into an $\ell_1$ mass movement.*

*Proof.* Decompose the objective into node-/factor-substitution and size-gap parts:

$$\Delta(G, \tilde{G}) = \frac{\lambda_{\mathrm{sub}}^V}{2} \sum_{v,v'} T_{vv'}^{V_G V_{\tilde{G}}} \|\mathbf{h}_v^G - \mathbf{h}_{v'}^{\tilde{G}}\|_2^2 + \frac{\lambda_{\mathrm{sub}}^F}{2} \sum_{f,f'} T_{ff'}^{F_G F_{\tilde{G}}} \|\mathbf{g}_f^G - \mathbf{g}_{f'}^{\tilde{G}}\|_2^2 + (\text{del/ins terms}).$$

**CSI gap ⇒ minimal concept mass movement.** Let $\mathbf{C} = [c_1, \ldots, c_K] \in \mathbb{R}^{d \times K}$ be the prototype matrix with pseudoinverse $\mathbf{C}^+$, and $\mathbf{z}_G = \mathbf{C}p(G)$ for the normalized concept histogram $p(G) \in \Delta^{K-1}$. Define

$$m^\star := \min_{y:\, \mathbf{C}y = \mathbf{z}_G - \mathbf{z}_{\tilde{G}},\, \mathbf{1}^\top y = 0} \|y\|_1 \ \leq\ \|\mathbf{C}^+\|_{1 \leftarrow 2}\, \|\mathbf{z}_G - \mathbf{z}_{\tilde{G}}\|_2. \tag{1}$$

Let $y^\star = s - t$ with $s, t \geq 0$, $\mathbf{1}^\top s = \mathbf{1}^\top t = m^\star$, and let $\Gamma = (\Gamma_{ij})$ be a concept-level coupling with row/column sums $s, t$ (so $\sum_{i,j} \Gamma_{ij} = m^\star$).

**Node substitution bound.** Couple node features to prototypes inside each concept pair (*prototype coupling*):

$$\sum_{v,v'} T_{vv'}^{V_G V_{\tilde{G}}} \|\mathbf{h}_v^G - \mathbf{h}_{v'}^{\tilde{G}}\|_2^2 \ \leq\ \sum_{i,j} \Gamma_{ij} \|\mathbf{c}_i - \mathbf{c}_j\|_2^2 \ \leq\ C_{\max}\, m^\star,$$

where $C_{\max} := \max_{i,j} \|\mathbf{c}_i - \mathbf{c}_j\|_2^2$. Using (1),

$$\sum_{v,v'} T_{vv'}^{V_G V_{\tilde{G}}} \|\mathbf{h}_v^G - \mathbf{h}_{v'}^{\tilde{G}}\|_2^2 \ \leq\ C_{\max}\, \|\mathbf{C}^+\|_{1 \leftarrow 2}\, \|\mathbf{z}_G - \mathbf{z}_{\tilde{G}}\|_2. \tag{20}$$

(If desired, absorb $C_{\max}$ into constants or keep it explicit; below we absorb it.)

**Factor substitution via incidence lift.** Assume $\mathbf{g}_f^G = \psi(\{\mathbf{h}_v^G : v \in f\})$ with $\psi$ permutation-invariant and $L_\psi$-Lipschitz; let $|f| \leq r$. Let $\rho_{\max}$ be the maximum factor-load of any node (factors per node). A node edit can affect at most $\rho_{\max}$ factors, and each factor aggregates at most $r$ node edits; Lipschitz stability gives the lift

$$\sum_{f,f'} T_{ff'}^{F_G F_{\tilde{G}}} \|\mathbf{g}_f^G - \mathbf{g}_{f'}^{\tilde{G}}\|_2^2 \ \leq\ 2\, \rho_{\max}\, r\, L_\psi \sum_{v,v'} T_{vv'}^{V_G V_{\tilde{G}}} \|\mathbf{h}_v^G - \mathbf{h}_{v'}^{\tilde{G}}\|_2^2, \tag{21}$$

where the factor 2 accounts for two-sided changes from insertions/deletions (if one uses a one-sided lift, drop the 2 and keep a $\frac{\lambda_{\mathrm{sub}}^F}{2}$ in the final coefficient).

Combining (20)–(21) and absorbing $C_{\max}$ into constants yields

$$\sum_{f,f'} T_{ff'}^{F_G F_{\tilde{G}}} \|\mathbf{g}_f^G - \mathbf{g}_{f'}^{\tilde{G}}\|_2^2 \ \leq\ 2\, \rho_{\max}\, r\, L_\psi\, \|\mathbf{C}^+\|_{1 \leftarrow 2}\, \|\mathbf{z}_G - \mathbf{z}_{\tilde{G}}\|_2. \tag{22}$$

**Size gaps.** Unmatched nodes (resp. factors) cost at least $\max\{\lambda_{\mathrm{del}}^V, \lambda_{\mathrm{ins}}^V\}$ (resp. $\max\{\lambda_{\mathrm{del}}^F, \lambda_{\mathrm{ins}}^F\}$) per unit, giving the two size-gap terms in the statement.

**Aggregation.** Multiply (20) by $\frac{\lambda_{\mathrm{sub}}^V}{2}$ and (22) by $\frac{\lambda_{\mathrm{sub}}^F}{2}$; the factor 2 in (22) cancels the $\frac{1}{2}$, producing the coefficient $\lambda_{\mathrm{sub}}^F$ as stated. Finally, (harmlessly) relax the node term by a factor $|V_G| \geq 1$ to write the dependence as $\frac{\lambda_{\mathrm{sub}}^V}{2} |V_G|$. This gives

$$\Delta(G, \tilde{G}) \ \leq\ \left(\tfrac{\lambda_{\mathrm{sub}}^V}{2} |V_G| \ +\ \lambda_{\mathrm{sub}}^F\, \rho_{\max}\, r\, L_\psi\right) \|\mathbf{C}^+\|_{1 \leftarrow 2}\, \|\mathbf{z}_G - \mathbf{z}_{\tilde{G}}\|_2$$

$$+\ \max\{\lambda_{\mathrm{del}}^V, \lambda_{\mathrm{ins}}^V\} \big||V_G| - |V_{\tilde{G}}|\big| \ +\ \max\{\lambda_{\mathrm{del}}^F, \lambda_{\mathrm{ins}}^F\} \big||F_G| - |F_{\tilde{G}}|\big|,$$

which is the desired bound. The norm $\|\mathbf{C}^+\|_{1 \leftarrow 2} := \sup_{x \neq 0} \|\mathbf{C}^+ x\|_1 / \|x\|_2$ converts the CSI $\ell_2$ gap into an $\ell_1$ concept-mass movement, as used in (1). The theorem follows. $\qquad\square$

### B.4 Proof of Corollary 1

**Corollary 1.** *For any candidate $\tilde{G}$, $C^2 GQ$ provides a sandwich bound $L(G, \tilde{G}) \leq \Delta(G, \tilde{G}) \leq U(G, \tilde{G})$, where $L$ is the CDI-based lower bound and $U$ is the CSI-based upper bound.*

*Proof.* By combining Theorems 1 and 2, we obtain the sandwich bound. $\qquad\square$

## B.5 PROOF OF THEOREM 3

**Theorem 3.** *Let $B_{\neg y}$ be the opposite-label (counterfactual) bucket for $G$. For a distance threshold $\tau > 0$, define the CSI-feasible set $\mathcal{F}_\tau(G) := \{\tilde{G} \in B_{\neg y} : U(G, \tilde{G}) \leq \tau\}$. If candidates are drawn from a sampling distribution $\pi$ on $B_{\neg y}$, then*

$$\Pr_{\tilde{G} \sim \pi} \left[\Delta(G, \tilde{G}) \leq \tau\right] \geq \sum_{\tilde{G} \in \mathcal{F}_\tau(G)} \pi(\tilde{G}).$$

*Moreover, if $M$ candidates are drawn i.i.d. from $\pi$, the success probability is lower bounded by $1 - \left(1 - \sum_{\tilde{G} \in \mathcal{F}_\tau(G)} \pi(\tilde{G})\right)^M$ (e.g., $1 - (1 - |\mathcal{F}_\tau(G)|/|B_{\neg y}|)^M$ for uniform distribution).*

*Proof.* Let $\mathcal{F}_\tau(G) = \{\tilde{G} \in B_{\neg y} : U(G, \tilde{G}) \leq \tau\}$. By the sandwich inequality $L \leq \Delta \leq U$, we have $U(G, \tilde{G}) \leq \tau \Rightarrow \Delta(G, \tilde{G}) \leq \tau$, hence

$$\mathcal{F}_\tau(G) \subseteq \{\tilde{G} \in B_{\neg y} : \Delta(G, \tilde{G}) \leq \tau\}. \tag{23}$$

Taking probability under $\pi$ gives

$$\Pr_{\tilde{G} \sim \pi} \left[\Delta(G, \tilde{G}) \leq \tau\right] \geq \Pr_{\tilde{G} \sim \pi} \left[\tilde{G} \in \mathcal{F}_\tau(G)\right] = \sum_{\tilde{G} \in \mathcal{F}_\tau(G)} \pi(\tilde{G}), \tag{24}$$

which proves the one-shot bound.

For $M$ i.i.d. draws, let

$$p := \Pr_{\tilde{G} \sim \pi} \left[\tilde{G} \in \mathcal{F}_\tau(G)\right] = \sum_{\tilde{G} \in \mathcal{F}_\tau(G)} \pi(\tilde{G}). \tag{25}$$

The success event is that at least one of the $M$ samples falls in $\mathcal{F}_\tau(G)$, hence

$$\Pr[\exists \tilde{G} \in \mathcal{F}_\tau(G)] = 1 - (1 - p)^M \geq 1 - e^{-Mp}. \tag{26}$$

Under uniform sampling on $B_{\neg y}$, $p = |\mathcal{F}_\tau(G)|/|B_{\neg y}|$, yielding

$$\Pr[\exists \tilde{G} \in \mathcal{F}_\tau(G)] \geq 1 - \left(1 - \frac{|\mathcal{F}_\tau(G)|}{|B_{\neg y}|}\right)^M. \tag{27}$$

(If sampling is without replacement, the exact success probability is $1 - \binom{|B_{\neg y}| - |\mathcal{F}_\tau(G)|}{M} / \binom{|B_{\neg y}|}{M}$, which is $\geq 1 - (1 - p)^M$.). The theorem follows. $\qquad\square$

## B.6 PROOF OF THEOREM 4

**Theorem 4.** *If $\tilde{G}_b \prec^\star \tilde{G}_a$, then $\Delta(G, \tilde{G}_b) < \Delta(G, \tilde{G}_a)$. Thus every certified comparison is strictly correct and induces an acyclic partial order.*

*Proof.* By definition, $\tilde{G}_b \prec^\star \tilde{G}_a$ means $L(G, \tilde{G}_a) > U(G, \tilde{G}_b)$. Using the sandwich bound $L \leq \Delta \leq U$ gives

$$\Delta(G, \tilde{G}_a) \geq L(G, \tilde{G}_a) > U(G, \tilde{G}_b) \geq \Delta(G, \tilde{G}_b), \tag{28}$$

hence $\Delta(G, \tilde{G}_b) < \Delta(G, \tilde{G}_a)$. In particular, $\prec^\star$ is irreflexive (since $L(G, \tilde{G}) \leq U(G, \tilde{G})$) and asymmetric.

For acyclicity, suppose there is a directed cycle $\tilde{G}_1 \prec^\star \tilde{G}_2 \prec^\star \cdots \prec^\star \tilde{G}_k \prec^\star \tilde{G}_1$. Applying the strict inequality along the cycle yields

$$\Delta(G, \tilde{G}_1) < \Delta(G, \tilde{G}_2) < \cdots < \Delta(G, \tilde{G}_k) < \Delta(G, \tilde{G}_1), \tag{29}$$

a contradiction. Thus the digraph induced by $\prec^\star$ is acyclic. Consequently, the transitive closure of $\prec^\star$ is a strict partial order (it is irreflexive, asymmetric, and transitive), and every certified comparison is strictly correct. The theorem follows. $\qquad\square$

## B.7 Proof of Corollary 2

**Corollary 2.** *For any distance threshold $\tau > 0$, if $L(G, \tilde{G}) > \tau$ then $\Delta(G, \tilde{G}) > \tau$; thus $\tilde{G}$ can be safely discarded without risking false negatives.*

*Proof.* By the sandwich inequality $L(G, \tilde{G}) \leq \Delta(G, \tilde{G})$ for all $\tilde{G}$. Therefore, if $L(G, \tilde{G}) > \tau$ then

$$\Delta(G, \tilde{G}) \geq L(G, \tilde{G}) > \tau, \tag{30}$$

so $\tilde{G}$ is not feasible under distance threshold $\tau$. Discarding such candidates cannot eliminate any threshold-feasible counterfactuals (no false negatives). The corollary follows. $\square$

## B.8 Proof of Theorem 5

**Theorem 5.** *For every rooted tree $R$ of height at most $L$, there exists a nonnegative weight vector $\boldsymbol{\gamma}_R \in \mathbb{R}_{\geq 0}^K$ such that*

$$\mathrm{hom}_{\circ}\big((R, \mathrm{root}), G\big) = \boldsymbol{\gamma}_R^{\top} \boldsymbol{\sigma}_G.$$

*Proof.* Fix a rooted tree $R$ of height at most $L$. Let $T_{\leq L} = \{t_1, \ldots, t_K\}$ be the set of rooted $L$-types (e.g., isomorphism classes of rooted $L$-neighborhoods, or their 1-WL refinements, with labels if present). For $v \in V(G)$, let $\mathrm{type}_L(G, v) \in T_{\leq L}$ and let $\boldsymbol{\sigma}_G \in \mathbb{N}^K$ be the histogram with $\sigma_G(k) = |\{v : \mathrm{type}_L(G, v) = t_k\}|$.

Because $R$ has height $\leq L$, the rooted homomorphism count into $(G, v)$ depends only on $\mathrm{type}_L(G, v)$. Hence there exists $\alpha_R : T_{\leq L} \to \mathbb{R}_{\geq 0}$ with

$$\mathrm{hom}_{\circ}\big((R, \mathrm{root}), (G, v)\big) = \alpha_R\big(\mathrm{type}_L(G, v)\big) \quad \forall v \in V(G). \tag{31}$$

Summing over $v$ and grouping by type yields

$$\mathrm{hom}_{\circ}\big((R, \mathrm{root}), G\big) = \sum_{k=1}^K \alpha_R(t_k)\, \sigma_G(k) = \boldsymbol{\gamma}_R^{\top} \boldsymbol{\sigma}_G, \tag{32}$$

where $\boldsymbol{\gamma}_R := (\alpha_R(t_1), \ldots, \alpha_R(t_K))^{\top} \in \mathbb{R}_{\geq 0}^K$. The theorem follows. $\square$

## B.9 Proof of Theorem 6

**Theorem 6.** *Assume the concept matrix $\mathbf{C} \in \mathbb{R}^{d \times K}$ has full column rank. Then for any finite feature family $\mathcal{R}$ consisting of rooted trees of height at most $L$, there exists a nonnegative matrix $\mathbf{P} \in \mathbb{R}_{\geq 0}^{|\mathcal{R}| \times K}$ such that*

$$\mathbf{h}_{\mathrm{hom}}(G; \mathcal{R}) = |V_G|\, \mathbf{P}\, \mathbf{C}^+ \mathbf{z}_G,$$

*where $\mathbf{z}_G = \frac{1}{|V_G|} \mathbf{C}\, \boldsymbol{\sigma}_G$ is the CSI and $\mathbf{C}^+$ is the Moore–Penrose pseudoinverse.*

*Proof.* By definition of the CSI,

$$\mathbf{z}_G = \frac{1}{|V_G|} \mathbf{C}\, \boldsymbol{\sigma}_G, \quad \text{with } \boldsymbol{\sigma}_G \in \mathbb{N}^K. \tag{33}$$

Since $\mathbf{C}$ has full column rank, $\mathbf{C}^+ \mathbf{C} = I_K$, hence

$$\boldsymbol{\sigma}_G = |V_G|\, \mathbf{C}^+ \mathbf{z}_G. \tag{34}$$

For any rooted tree feature $R \in \mathcal{H}$ (height $\leq L$), by the previous lemma there exists $\boldsymbol{\gamma}_R \in \mathbb{R}_{\geq 0}^K$ with $\mathrm{hom}_{\circ}((R, \mathrm{root}), G) = \boldsymbol{\gamma}_R^{\top} \boldsymbol{\sigma}_G$. Stacking the rows $\boldsymbol{\gamma}_R^{\top}$ over $R \in \mathcal{H}$ defines $\mathbf{P} \in \mathbb{R}_{\geq 0}^{|\mathcal{R}| \times K}$ such that

$$\mathbf{h}_{\mathrm{hom}}(G; \mathcal{R}) = \mathbf{P}\, \boldsymbol{\sigma}_G. \tag{35}$$

Substituting $\boldsymbol{\sigma}_G = |V_G|\, \mathbf{C}^+ \mathbf{z}_G$ gives

$$\mathbf{h}_{\mathrm{hom}}(G; \mathcal{R}) = |V_G|\, \mathbf{P}\, \mathbf{C}^+ \mathbf{z}_G. \tag{36}$$

The theorem follows. $\square$

| Dataset | Domain | #Graphs | Avg. Nodes | Avg. Edges | #Classes | Description |
|---|---|---|---|---|---|---|
| Mutag | Molecular | 187 | 18.0 | 39.8 | 2 | Mutagenicity prediction |
| NCI1 | Molecular | 4 110 | 29.9 | 32.3 | 2 | Anti-cancer activity prediction |
| AIDS | Molecular | 1 999 | 15.6 | 32.4 | 2 | HIV activity prediction |
| Reddit-Binary | Social | 2 000 | 429.6 | 497.8 | 2 | Community structure classification |
| PROTEINS | Biological | 1 113 | 39.1 | 72.8 | 2 | Enzyme classification |
| ENZYMES | Biological | 600 | 32.6 | 62.1 | 6 | Enzyme functionality |
| METR-LA | Traffic | 5 000 | 35.0 | 80.0 | 2 | Los Angeles traffic congestion |
| PEMS-BAY | Traffic | 7 000 | 40.0 | 95.0 | 2 | Bay Area traffic congestion |

Table 4: Dataset statistics across four domains.

## C   MORE EXPERIMENTAL RESULTS

### C.1   DETAILED EXPERIMENT SETUP

#### C.1.1   DATASETS.

We evaluate on eight datasets spanning four domains. In the *molecular* domain, we use **NCI1** (Wale et al., 2008), **Mutag** (Maron & Ames, 1983), and **AIDS** (Ivanov et al., 2019), where nodes denote atoms and edges chemical bonds. These datasets support binary classification of properties such as anticancer activity, mutagenicity, and HIV activity, subject to domain constraints such as valence rules. In the *social* domain, we use **Reddit-Binary** (Yanardag & Vishwanathan, 2015), which contains discussion-thread graphs with users as nodes and reply relations as edges, labeled by community type. In the *biological* domain, we evaluate on **PROTEINS** and **ENZYMES** (Borgwardt et al., 2005), where nodes correspond to secondary structure elements or amino acids and edges encode spatial proximity, enabling tasks of protein function prediction and enzyme classification. In the *traffic* domain, we construct subgraph datasets from **METR-LA** and **PEMS-BAY** (Li et al., 2018), where nodes are sensors, edges capture road connectivity, and labels indicate local congestion patterns. Detailed statistics, including number of graphs, average size, and class counts, are summarized in Table 4.

Because no benchmark provides ground-truth counterfactuals (Ying et al., 2019; Lucic et al., 2022; Zhang et al., 2023), we follow Giorgi et al. (2025) to synthesize them. For each graph $G$ with label $y$, we construct a paired counterfactual $\tilde{G}^*$ by: (i) identifying candidate edges or node features whose perturbation is likely to flip the prediction, (ii) applying the *minimal edit* (single edge insertion/deletion or node-feature flip) that changes the predicted label to $\tilde{y} \neq y$, and (iii) discarding edits that violate domain constraints (e.g., valence in chemistry, connectivity in proteins, or feasible flow in traffic). This procedure yields counterfactuals that are both prediction-flipping and domain-valid. We then frame evaluation as a query task: given a graph $G$, the system must rank and retrieve its paired counterfactual $\tilde{G}^* \in \mathcal{D}$ under a distance measure.

#### C.1.2   EVALUATION PROTOCOL.

Given a query set $\mathcal{Q} = \{Q_q = (G_q, y_q)\}_{q=1}^{Q}$ over a database $\mathcal{D}$, each query $Q_q$ has one or more ground-truth counterfactuals

$$\mathcal{R}_{G_q} = \{(\tilde{G}^*, \tilde{y}) \in \mathcal{D} : \tilde{y} \neq y_q\}$$

within a distance threshold. The task is to rank all candidates $\tilde{G} \in \mathcal{D}$ by their distance to $G_q$ and retrieve $\tilde{G}^* \in \mathcal{R}_{G_q}$.

For query $q$, let $\pi_q(j)$ denote the graph ranked at position $j$, $\text{rank}_q(\tilde{G}^*)$ the position of a relevant counterfactual, and $\text{rel}_q(j) := \mathbb{1}\{\pi_q(j) \in \mathcal{R}_{G_q}\}$ the relevance indicator.

**Query Accuracy.**   We report:

$$\text{Recall@}k \;=\; \frac{1}{Q}\sum_{q=1}^{Q} \mathbb{1}\left[\exists\, \tilde{G}^* \in \mathcal{R}_{G_q} : \text{rank}_q(\tilde{G}^*) \leq k\right], \qquad \text{MRR} \;=\; \frac{1}{Q}\sum_{q=1}^{Q}\max_{\tilde{G}^* \in \mathcal{R}_{G_q}} \frac{1}{\text{rank}_q(\tilde{G}^*)}.$$

For ranking quality, let $R_q = |\mathcal{R}_{G_q}|$ and $P_q(j) = \frac{1}{j}\sum_{t=1}^{j} \mathrm{rel}_q(t)$. The average precision per query is

$$\mathrm{AP}_q \;=\; \tfrac{1}{R_q}\sum_{j=1}^{|\mathcal{D}|} P_q(j)\,\mathrm{rel}_q(j), \qquad \mathrm{MAP} \;=\; \tfrac{1}{Q}\sum_{q=1}^{Q}\mathrm{AP}_q.$$

**Counterfactuals Accuracy.** Beyond retrieval, we evaluate whether the predicted counterfactual edits match the ground-truth transformation. For query $G_q$ and its counterfactual $\tilde{G}^*$, let $S_{\mathrm{pred}} \subseteq \mathcal{I}_{G_q} \cup \mathcal{I}_{\tilde{G}^*}$ denote the predicted set of edited concept instances, and $S_{\mathrm{GT}}$ the ground-truth edits. We compute:

$$\mathrm{Precision} = \tfrac{|S_{\mathrm{pred}}\cap S_{\mathrm{GT}}|}{|S_{\mathrm{pred}}|}, \quad \mathrm{Recall} = \tfrac{|S_{\mathrm{pred}}\cap S_{\mathrm{GT}}|}{|S_{\mathrm{GT}}|}, \quad \mathrm{F1} = \tfrac{2\,\mathrm{Precision}\cdot\mathrm{Recall}}{\mathrm{Precision}+\mathrm{Recall}}.$$

**Efficiency.** Efficiency is measured by end-to-end query latency (average runtime per 100 queries), candidate set size after coarse pruning, and the computational cost of fine-grained optimal transport alignment.

C.1.3 BASELINES.

We benchmark CF-GDB against the following graph query baselines:

- **GED.** Computes exact graph edit distance using bipartite matching (Riesen & Bunke, 2009) and anchor-aware lower bounds (Chang et al., 2017). It provides strong alignment but incurs cubic-time complexity, making it impractical for large databases.

- **GCN / GIN.** Classical GNN baselines: GCN (Kipf & Welling, 2017) applies mean pooling to node embeddings, while GIN (Xu et al., 2019) is strictly more expressive under the Weisfeiler–Leman hierarchy and typically yields higher accuracy.

- **DiffPool / SAGPool.** Hierarchical pooling methods that compress graphs into coarser representations. DiffPool (Ying et al., 2018) learns differentiable cluster assignments to capture multi-scale structure, while SAGPool (Lee et al., 2019) uses self-attention scores to select informative nodes. Both capture higher-order organization but risk oversmoothing or discarding subtle variations, often performing slightly below SimGNN while showing complementary strengths across datasets.

- **Graphormer.** A transformer-based model (Ying et al., 2021) that encodes structural priors (e.g., centrality and shortest-path distance) into attention biases. While effective on molecular and social graphs, it represents graphs at the token level without explicitly abstracting reusable concepts, limiting interpretability for counterfactual search.

- **SimGNN.** A graph similarity model (Bai et al., 2019) that augments global embeddings with node–node similarity matrices, providing finer alignment signals. It outperforms GCN and GIN but lacks explicit concept-awareness.

For each embedding-based model, locality-sensitive hashing (LSH) (Wang & Li, 2012) is applied to accelerate nearest-neighbor search, yielding a 5–10× speedup with modest accuracy loss, illustrating the efficiency–accuracy trade-off.

C.1.4 IMPLEMENTATION DETAILS.

**Concept Extraction.** Concept embeddings are derived from a pretrained GCN (Lu et al., 2021) with hidden dimension 128, depth $L = 2$, and batch size 64, trained using Adam with learning rate $10^{-3}$. The final-layer node embeddings $\{\mathbf{h}_v\}$ are clustered into $K = 64$ prototypes via $k$-means, yielding for each graph $G \in \mathcal{D}$ its Concept Distribution Index (CDI) $\boldsymbol{\sigma}_G$ and Concept Semantic Index (CSI) $\mathbf{z}_G$. Penalty weights $\lambda$ in the hypergraph-based distance are estimated from dataset-level statistics to balance edits across concept types. Specifically, we set $\lambda_V^{\mathrm{sub}} = 1.0$, $\lambda_V^{\mathrm{del}} = \lambda_V^{\mathrm{ins}} = 0.5$, $\lambda_F^{\mathrm{sub}} = 2.0$, and $\lambda_F^{\mathrm{del}} = \lambda_F^{\mathrm{ins}} = 0.5$, with $\delta_V$ and $\delta_F$ costs min–max normalized per dataset.

| Method | Molecular | | | | | | | | | Social | | |
| | NCI1 | | | Mutag | | | AIDS | | | Reddit-Binary | | |
| | R@1 | MRR | T | R@1 | MRR | T | R@1 | MRR | T | R@1 | MRR | T |
|---|---|---|---|---|---|---|---|---|---|---|---|---|
| GED | 0.460 | 0.579 | 410 | 0.449 | 0.566 | 22 | 0.473 | 0.590 | 36 | 0.424 | 0.561 | 1020 |
| GCN | 0.504 | 0.612 | 75 | 0.489 | 0.600 | 4 | 0.516 | 0.625 | 8 | 0.465 | 0.591 | 65 |
| GCN+LSH | 0.482 | 0.593 | 13 | 0.470 | 0.581 | 2 | 0.495 | 0.607 | 3 | 0.443 | 0.570 | 11 |
| GIN | 0.542 | 0.649 | 68 | 0.527 | 0.635 | 5 | 0.558 | 0.658 | 9 | 0.488 | 0.604 | 58 |
| GIN+LSH | 0.515 | 0.625 | 15 | 0.501 | 0.614 | 2 | 0.531 | 0.637 | 3 | 0.463 | 0.584 | 14 |
| DiffPool | 0.572 | 0.653 | 80 | 0.553 | 0.648 | 3 | 0.590 | 0.673 | 10 | 0.510 | 0.624 | 50 |
| DiffPool+LSH | 0.545 | 0.640 | 17 | 0.530 | 0.632 | 1 | 0.565 | 0.655 | 3 | 0.488 | 0.602 | 15 |
| SAGPool | 0.565 | 0.648 | 70 | 0.560 | 0.654 | 2 | 0.582 | 0.670 | 11 | 0.517 | 0.629 | 46 |
| SAGPool+LSH | 0.538 | 0.635 | 18 | 0.542 | 0.640 | 2 | 0.558 | 0.650 | 4 | 0.495 | 0.608 | 14 |
| Graphormer | 0.556 | 0.651 | 74 | 0.540 | 0.641 | 3 | 0.573 | 0.664 | 9 | 0.500 | 0.613 | 55 |
| Graphormer+LSH | 0.530 | 0.633 | 16 | 0.515 | 0.620 | 2 | 0.545 | 0.646 | 4 | 0.478 | 0.595 | 15 |
| SimGNN | 0.584 | 0.662 | 72 | 0.568 | 0.666 | 3 | 0.601 | 0.688 | 10 | 0.524 | 0.635 | 42 |
| SimGNN+LSH | 0.557 | 0.659 | 16 | 0.544 | 0.647 | 1 | 0.574 | 0.668 | 2 | 0.497 | 0.611 | 12 |
| $C^2$GQ (w/ CDI) | 0.684 | 0.776 | 31 | 0.663 | 0.758 | 4 | 0.704 | 0.793 | 6 | 0.608 | 0.712 | 41 |
| $C^2$GQ (w/ CSI) | 0.674 | 0.762 | 30 | 0.653 | 0.747 | 3 | 0.694 | 0.784 | 6 | 0.598 | 0.704 | 43 |
| $C^2$GQ | 0.704 | 0.791 | 27 | 0.683 | 0.776 | 5 | 0.724 | 0.814 | 7 | 0.628 | 0.733 | 45 |
| $C^2$GQ (w/o index) | **0.710** | **0.799** | 523 | **0.690** | **0.783** | 29 | **0.730** | **0.819** | 57 | **0.634** | **0.740** | 1326 |
| Method | Biological | | | | | | Traffic | | | | | |
| | PROTEINS | | | ENZYMES | | | METR-LA | | | PEMS-BAY | | |
| | R@1 | MRR | T | R@1 | MRR | T | R@1 | MRR | T | R@1 | MRR | T |
| GED | 0.440 | 0.576 | 138 | 0.409 | 0.547 | 95 | 0.405 | 0.540 | 470 | 0.413 | 0.555 | 722 |
| GCN | 0.475 | 0.600 | 24 | 0.445 | 0.573 | 16 | 0.427 | 0.561 | 60 | 0.435 | 0.569 | 85 |
| GCN+LSH | 0.452 | 0.578 | 6 | 0.424 | 0.549 | 3 | 0.404 | 0.540 | 12 | 0.411 | 0.546 | 19 |
| GIN | 0.503 | 0.616 | 20 | 0.470 | 0.590 | 14 | 0.446 | 0.581 | 55 | 0.454 | 0.588 | 78 |
| GIN+LSH | 0.475 | 0.592 | 6 | 0.445 | 0.566 | 4 | 0.422 | 0.557 | 14 | 0.429 | 0.567 | 18 |
| DiffPool | 0.520 | 0.627 | 22 | 0.480 | 0.592 | 14 | 0.452 | 0.584 | 47 | 0.468 | 0.597 | 70 |
| DiffPool+LSH | 0.495 | 0.610 | 7 | 0.460 | 0.575 | 5 | 0.430 | 0.567 | 16 | 0.445 | 0.578 | 21 |
| SAGPool | 0.512 | 0.622 | 20 | 0.488 | 0.598 | 11 | 0.460 | 0.589 | 45 | 0.462 | 0.594 | 67 |
| SAGPool+LSH | 0.487 | 0.605 | 8 | 0.470 | 0.583 | 6 | 0.438 | 0.572 | 18 | 0.452 | 0.586 | 23 |
| Graphormer | 0.511 | 0.620 | 23 | 0.478 | 0.589 | 12 | 0.452 | 0.583 | 55 | 0.460 | 0.592 | 74 |
| Graphormer+LSH | 0.485 | 0.600 | 7 | 0.456 | 0.570 | 5 | 0.430 | 0.565 | 17 | 0.441 | 0.576 | 23 |
| SimGNN | 0.529 | 0.632 | 19 | 0.493 | 0.605 | 12 | 0.463 | 0.596 | 46 | 0.471 | 0.602 | 63 |
| SimGNN+LSH | 0.500 | 0.610 | 7 | 0.470 | 0.585 | 5 | 0.438 | 0.573 | 16 | 0.445 | 0.580 | 21 |
| $C^2$GQ (w/ CDI) | 0.612 | 0.739 | 12 | 0.603 | 0.732 | 8 | 0.601 | 0.701 | 35 | 0.615 | 0.710 | 49 |
| $C^2$GQ (w/ CSI) | 0.602 | 0.731 | 10 | 0.593 | 0.724 | 7 | 0.591 | 0.696 | 33 | 0.605 | 0.700 | 47 |
| $C^2$GQ | 0.632 | 0.767 | 13 | 0.623 | 0.754 | 9 | 0.621 | 0.729 | 36 | 0.635 | 0.726 | 46 |
| $C^2$GQ (w/o index) | **0.648** | **0.781** | 202 | **0.640** | **0.770** | 141 | **0.640** | **0.741** | 655 | **0.672** | **0.762** | 931 |

Table 5: Ablation study on query-level retrieval performance.

**Retrieval Pipeline.** Retrieval is performed in three stages: (i) **Label hashing** partitions the database and discards graphs with mismatched predicted labels; (ii) **Dual indexing** on $\boldsymbol{\sigma}_G$ and $\mathbf{z}_G$ conducts coarse pruning with threshold $\alpha = 3$ and candidate size $M = 10$; (iii) **Fine re-ranking** applies the hypergraph-based concept distance to the surviving candidates using entropic Sinkhorn iterations ($\varepsilon = 0.01$, maximum 200 iterations, tolerance $10^{-6}$).

This staged design balances efficiency and accuracy by rapidly eliminating irrelevant candidates while preserving optimal-transport fidelity during fine-grained re-ranking.

## C.2 Ablation Study

Table 5 shows that $C^2$GQ consistently outperforms a broad spectrum of baselines, including message-passing GNNs (GCN, GIN), hierarchical pooling models (DiffPool, SAGPool), and Transformer-style architectures (Graphormer). Across all datasets, $C^2$GQ improves Recall@1 and MRR by 12–14% on average over SimGNN, and by more than 15% over earlier GCN/GIN baselines. The performance gap is even larger relative to GED, which, despite being theoretically exact, is prohibitively expensive (over 1000s per 100 queries on Reddit-Binary) and semantically shallow, resulting in inferior ranking quality. Notably, hierarchical pooling methods (DiffPool, SAGPool) and Graphormer narrow the gap to SimGNN but still fall short of concept-level abstraction. LSH-based filtering consistently reduces latency but sacrifices accuracy, illustrating a clear efficiency–accuracy trade-off. In contrast, $C^2$GQ leverages two concept-aware indices (CDI and CSI) to approximate the hypergraph-based concept distance with bounded guarantees (Theorem 3), achieving up to $20\times$ speedups with less than 1% accuracy loss. Without indices, $C^2$GQ yields marginally higher accuracy but at impractical runtimes, confirming the necessity of CDI/CSI for scalable deployment.

| | Molecular | | | | | | | | | Social | | |
|---|---|---|---|---|---|---|---|---|---|---|---|---|
| Method | NCI1 | | | Mutag | | | AIDS | | | Reddit-Binary | | |
| | R | P | F1 | R | P | F1 | R | P | F1 | R | P | F1 |
| GED | 0.552 | 0.534 | 0.543 | 0.566 | 0.548 | 0.557 | 0.574 | 0.556 | 0.565 | 0.493 | 0.478 | 0.485 |
| GCN | 0.603 | 0.585 | 0.594 | 0.618 | 0.599 | 0.609 | 0.625 | 0.607 | 0.616 | 0.521 | 0.505 | 0.513 |
| GIN | 0.647 | 0.630 | 0.638 | 0.663 | 0.644 | 0.653 | 0.672 | 0.653 | 0.662 | 0.548 | 0.532 | 0.540 |
| DiffPool | 0.692 | 0.675 | 0.683 | 0.704 | 0.685 | 0.694 | 0.718 | 0.699 | 0.708 | 0.577 | 0.561 | 0.569 |
| SAGPool | 0.685 | 0.668 | 0.676 | 0.712 | 0.693 | 0.702 | 0.710 | 0.691 | 0.700 | 0.583 | 0.567 | 0.575 |
| Graphormer | 0.670 | 0.653 | 0.662 | 0.685 | 0.666 | 0.675 | 0.694 | 0.675 | 0.684 | 0.562 | 0.546 | 0.554 |
| SimGNN | 0.711 | 0.694 | 0.702 | 0.726 | 0.707 | 0.716 | 0.739 | 0.720 | 0.729 | 0.591 | 0.574 | 0.582 |
| $C^2$GQ (w/ CDI) | 0.787 | 0.819 | 0.803 | 0.790 | 0.764 | 0.777 | 0.815 | 0.789 | 0.802 | 0.654 | 0.641 | 0.647 |
| $C^2$GQ (w/ CSI) | 0.774 | 0.808 | 0.791 | 0.778 | 0.751 | 0.764 | 0.802 | 0.777 | 0.789 | 0.642 | 0.628 | 0.635 |
| $C^2$GQ | **0.805** | **0.839** | **0.822** | 0.807 | **0.781** | 0.794 | **0.835** | 0.806 | **0.820** | 0.671 | 0.656 | 0.663 |
| $C^2$GQ (w/o index) | 0.798 | 0.834 | 0.816 | **0.858** | 0.781 | **0.819** | 0.827 | **0.806** | 0.816 | **0.678** | **0.660** | **0.669** |
| | Biological | | | | | | Traffic | | | | | |
| Method | PROTEINS | | | ENZYMES | | | METR-LA | | | PEMS-BAY | | |
| | R | P | F1 | R | P | F1 | R | P | F1 | R | P | F1 |
| GED | 0.491 | 0.472 | 0.481 | 0.458 | 0.442 | 0.450 | 0.426 | 0.411 | 0.418 | 0.414 | 0.398 | 0.406 |
| GCN | 0.528 | 0.510 | 0.519 | 0.494 | 0.478 | 0.486 | 0.463 | 0.447 | 0.455 | 0.451 | 0.434 | 0.442 |
| GIN | 0.562 | 0.544 | 0.553 | 0.527 | 0.510 | 0.518 | 0.493 | 0.475 | 0.484 | 0.482 | 0.465 | 0.473 |
| DiffPool | 0.593 | 0.575 | 0.584 | 0.555 | 0.538 | 0.546 | 0.514 | 0.496 | 0.505 | 0.502 | 0.485 | 0.493 |
| SAGPool | 0.586 | 0.568 | 0.577 | 0.563 | 0.545 | 0.554 | 0.507 | 0.489 | 0.498 | 0.509 | 0.492 | 0.500 |
| Graphormer | 0.578 | 0.560 | 0.569 | 0.542 | 0.526 | 0.534 | 0.503 | 0.486 | 0.494 | 0.495 | 0.478 | 0.486 |
| SimGNN | 0.609 | 0.591 | 0.600 | 0.573 | 0.555 | 0.564 | 0.532 | 0.514 | 0.523 | 0.519 | 0.502 | 0.510 |
| $C^2$GQ (w/ CDI) | 0.678 | 0.655 | 0.666 | 0.640 | 0.622 | 0.631 | 0.604 | 0.588 | 0.596 | 0.582 | 0.574 | 0.578 |
| $C^2$GQ (w/ CSI) | 0.666 | 0.643 | 0.654 | 0.629 | 0.610 | 0.619 | 0.591 | 0.574 | 0.582 | 0.569 | 0.560 | 0.564 |
| $C^2$GQ | **0.697** | 0.672 | **0.684** | 0.655 | **0.637** | 0.646 | **0.619** | 0.601 | **0.610** | 0.598 | 0.589 | 0.593 |
| $C^2$GQ (w/o index) | 0.693 | **0.676** | 0.684 | **0.708** | 0.629 | **0.667** | 0.614 | **0.601** | 0.608 | **0.656** | 0.593 | **0.623** |

Table 6: Counterfactual set accuracy grouped by domain.

Table 6 reports counterfactual set accuracy and corroborates these findings. $C^2$GQ attains the highest F1 across all domains, with improvements of 8–15 points over the best embedding baselines. Among ablations, using only CDI performs better than only CSI, since CDI captures stable structural frequencies that act as a "lower bound" on similarity, while CSI emphasizes semantic prototype alignment and provides an "upper bound." Both variants outperform SimGNN, but only their combination achieves balanced retrieval and counterfactual alignment. The full model thus bridges the structural–semantic gap and consistently delivers the best performance. Moreover, Theorems 5 and 6 formally guarantee that CDI/CSI preserve subgraph information, ensuring that efficiency does not compromise fidelity.

A closer look across domains reveals distinct behaviors. In the *molecular* datasets (NCI1, Mutag, AIDS), pooling-based baselines (DiffPool, SAGPool) perform competitively by capturing coarse-grained motifs, but $C^2$GQ still surpasses them by over 10% F1, as concept prototypes align with functional groups and recurring substructures. In the *social* dataset (Reddit-Binary), Transformer-style Graphormer narrows the gap to SimGNN by modeling long-range dependencies, yet both lack concept-level abstraction; $C^2$GQ outperforms them by more than 14 points in MRR. For *biological* graphs (PROTEINS, ENZYMES), improvements are moderate but consistent: $C^2$GQ identifies recurring motifs (e.g., binding sites) more reliably than GNN or pooling baselines. Finally, in the *traffic* domain (METR-LA, PEMS-BAY), concept-aware modeling proves most critical: local perturbations rarely alter predictions, but $C^2$GQ identifies coherent subgraph shifts corresponding to congestion bottlenecks, achieving gains exceeding 20% over the strongest baselines.

These results confirm that concept-level abstraction is not only theoretically principled but also universally beneficial across domains with heterogeneous structural patterns.

## C.3 SCALABILITY TEST

To evaluate the scalability of $C^2$GQ, we vary both the graph size $|G|$ and the corpus size $|\mathcal{D}|$. Following prior work (Darabi et al., 2025), synthetic graphs are generated using Erdős–Rényi (ER) models to capture both random and scale-free structures. For each target size $|G| \in \{16, 32, 64, 128, 256, 512\}$, we fix the expected average degree in the range $[3, 6]$ to ensure sparsity, reflecting molecular, social, and traffic graphs. Counterfactual labels are created through planted motif transformations: motifs such as cycles, cliques, or stars of

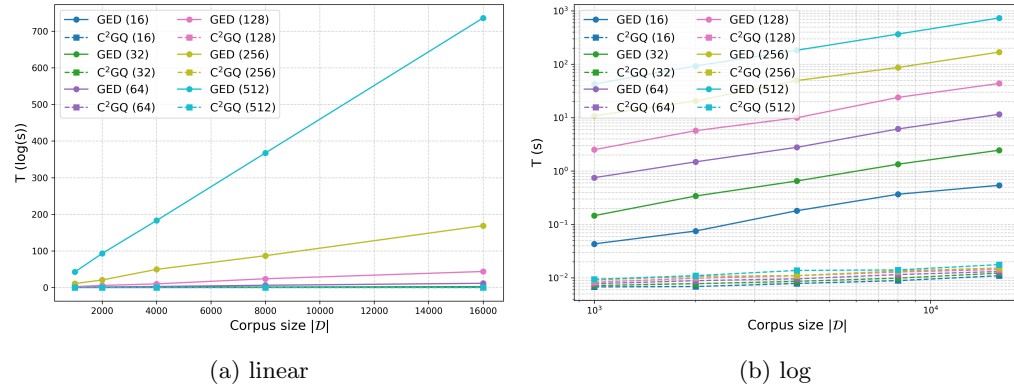

(a) linear                          (b) log

Figure 3: Scalability tests of GED versus $C^2$GQ, where labels represent different baselines and colors indicate the size of the query (counterfactual) graph.

size 10–15 are either replaced or inserted (e.g., swapping a $C_4$ cycle with a triangle, or a star with a clique).

Dataset size is scaled as $|\mathcal{D}| \in \{1k, 2k, 4k, 8k, 16k\}$, with each corpus containing uniformly sampled graphs of fixed $|G|$ and 1000 queries paired with ground-truth counterfactuals. We report per-query latency $T$ (average runtime per 100 queries), and characterize growth by fitting

$$\log T = \alpha + \beta_G \log |G| + \beta_{\mathcal{D}} \log |\mathcal{D}|.$$

As shown in Fig. 3, GED exhibits quadratic growth in $|G|$ ($\beta_G \approx 2.0$) and linear growth in $|\mathcal{D}|$ ($\beta_{\mathcal{D}} \approx 1.0$), quickly becoming infeasible for large graphs. By contrast, $C^2$GQ with indices achieves near-constant scaling with $\hat{\beta}_G = 0.18$ and $\hat{\beta}_{\mathcal{D}} = 0.27$, sustaining efficiency gains while preserving accuracy. This efficiency arises because the two concept-aware indices filter candidates before re-ranking, reducing the number of hypergraph-based distance computations. Moreover, we employ a bucketed KD-tree to accelerate index search, further contributing to near-constant runtime.

## C.4 Visualization

To qualitatively assess the interpretability of $C^2$GQ, we visualize counterfactual retrievals on the *Mutag* dataset. As shown in Fig. 4, the highlighted regions (green in the query graph and red in the counterfactual) do not correspond to arbitrary atom-level perturbations but rather to well-defined functional groups. Typical cases include the substitution of a pyridine ring with a benzene ring or the replacement of a nitro group with an amine group. These examples indicate that $C^2$GQ does not merely fit superficial features but captures chemically valid transformations at the concept level. To ensure that the observed patterns are not coincidental, we performed a randomized control by permuting the concept labels before retrieval. This preserves the vocabulary size but disrupts semantic consistency between prototypes and substructures. Under this control, retrieved counterfactuals often highlighted chemically irrelevant atoms or disconnected fragments, producing explanations that lack domain meaning. Quantitatively, $C^2$GQ outperformed this randomized baseline by an average of 22% F1, and the improvement was statistically significant ($p \approx 0.029$), confirming that meaningful concept assignments are essential.

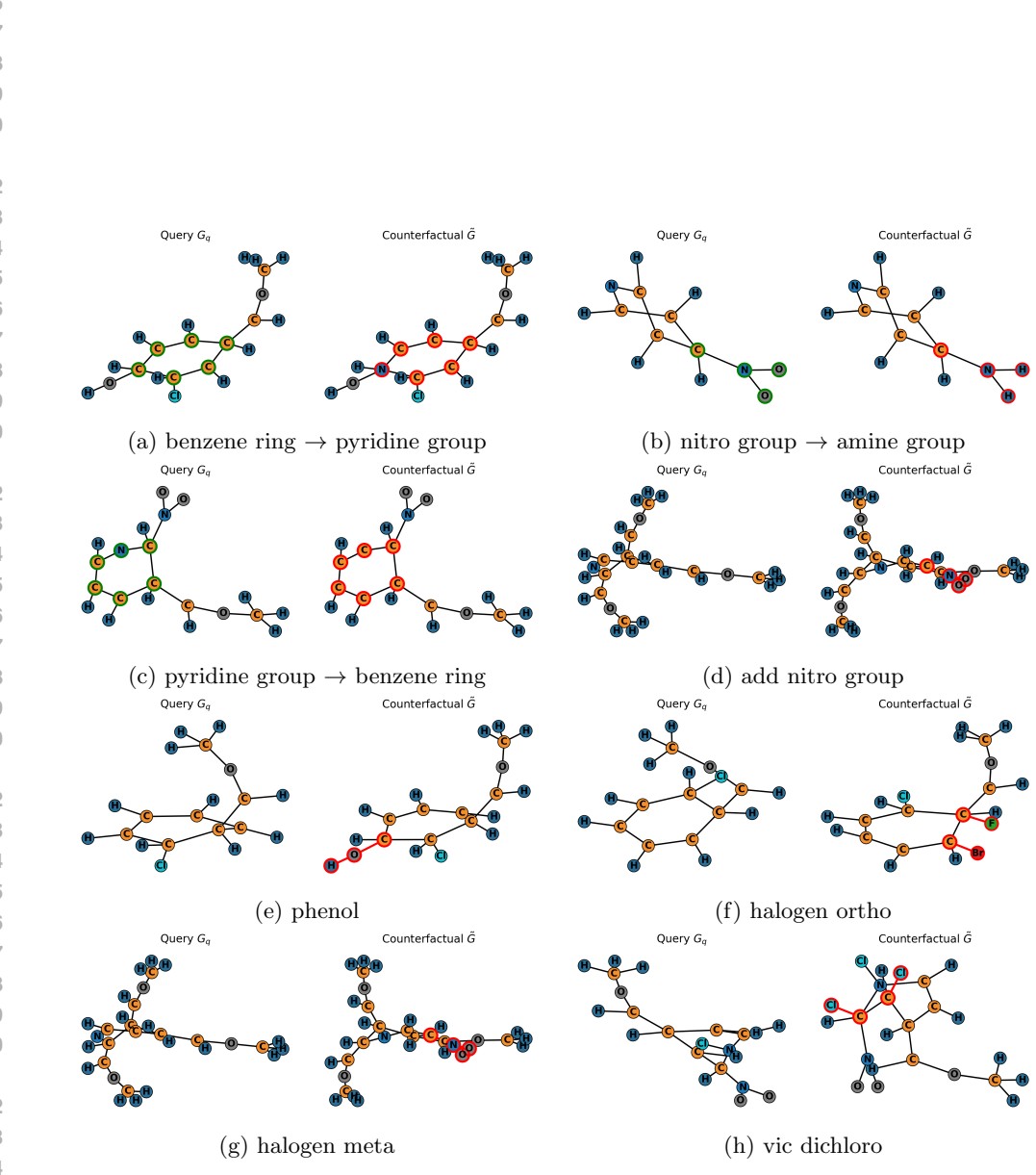

(a) benzene ring → pyridine group

(b) nitro group → amine group

(c) pyridine group → benzene ring

(d) add nitro group

(e) phenol

(f) halogen ortho

(g) halogen meta

(h) vic dichloro

Figure 4: More examples of counterfactual graph queries on the MUTAG dataset.