# OpenReview forum: "Answering Counterfactual Queries on Graph Databases"
_ICLR.cc/2026/Conference — ICLR 2026 Conference Desk Rejected Submission_

### Official Review · Reviewer_MQnh · 2025-10-31

**Soundness:** 2
**Presentation:** 3
**Contribution:** 2
**Rating:** 4
**Confidence:** 2

**Summary:**

In this submission, the author(s)  focus(es) on the problem of graph query on counterfactual datasets. In particular, a query-based framework called the Counterfactual Graph Database (CF-GDB) has been proposed to enable counterfactual reasoning on graphs. There are two more modules: the Concept Distribution Index (CDI) and the Concept Semantic Index (CSI). These two modules can further efficiency and scalability of the proposed method. Extensive experiments show the effectiveness of the CF-GDB framework.

**Strengths:**

This submission has the following strengths:
- This paper is clearly motivated and well organized.
- This is the first work on retrieving the most relevant counterfacual graphs.
- The paper provides detailed theoretical analysis.
- The experimental results demonstrate the effectiveness of the proposed framework.

**Weaknesses:**

This submission has the following weaknesses:
- This work might be closely related to graph retrieval-augmented generation. The high-level idea is similar. The difference is that the query is graph and the database is counterfactuals.
- There are no experiments on downstream tasks using the proposed framework.

**Questions:**

I have the following questions/suggestions:
- Are there existing studies that retrieve relevant counterfactual samples from databases rather than graphs?
- It would be better to discuss the differences compared to graph retrieval-augmented generation.

---

> ### Author Response · Authors · 2025-11-20
> **Reply of W1/Q2 [Comparison with GraphRAG]**
>
> #### Table A. Comparison between Graph RAG and CF-GDB
> | Aspect                         | **CF-GDB**                                                                                                                                                 | **Graph RAG** [1-6]                                                                                                        |
> | - | - | - |
> | Goal                     | Retrieve semantic counterfactual graphs in the database that flip the predicted label and reveal the underlying concepts (subgraphs) driving the decision. | Retrieve knowledge via graph-structured representations rather than flat text to improve question answering and reasoning. |
> | Focus                   | Interpretability through decision-boundary analysis                                                                                                        | Improving downstream QA, reasoning, or generation                                                                          |
> | Query               | A target graph whose predicted label is to be flipped                                                                                                      | A natural-language question or reasoning query                                                                             |
> | Database         | A database of real graphs (molecular, traffic, biological, social)                                                                                         | Text corpus + knowledge graph / entity graph                                                                               |
> | Output                     | Counterfactual graphs whose labels differ from the query but remain semantically similar                                                                                            | Relevant entities, subgraphs, or graph-structured text chunks                                                              |
> | Retrieval Criterion        | Concept-distribution divergence + OT-based semantic distance                                                                                               | Graph/embedding-based similarity for QA relevance                                                                          |
> | Interpretability           | Identifies which subgraphs cause prediction flips                                                                                                          | Not primarily designed for model explanation                                                                               |
> | Domain                     | Molecular, social, biological, traffic graphs                                                                                                              | Knowledge graphs for QA and multi-hop reasoning                                                                            |
>
> ### Goal of Graph RAG
> Sorry for misunderstanding. **GraphRAG [1–6] and our counterfactual graph query (CF-GDB) solve different tasks.** GraphRAG takes a *natural-language query* and retrieves text-derived KG subgraphs as *supportive evidence* for QA or reasoning. In contrast, **CF-GDB takes a graph as input** and retrieves *existing graphs* that **flip the model’s prediction**, requiring **contrastive, decision-boundary–aware retrieval** rather than similarity-based relevance. Since GraphRAG optimizes for embedding similarity, its mechanism cannot perform prediction-flipping counterfactual search. Table A summarizes the key differences.
>
> ### Key methodological differences
> - **Retrieval objective:** GraphRAG retrieves graphs similar to the query in embedding space, but counterfactuals lie across the decision boundary and are often structurally dissimilar. Similarity-based retrieval therefore cannot surface prediction-flipping candidates.
> - **Representation:** GraphRAG embeddings capture semantic similarity, not decision influence. CF-GDB instead uses CDI/CSI and an OT distance tied to the classifier, ensuring retrieval aligns with prediction changes.
> - **Evidence role:** GraphRAG retrieves *supportive* evidence, whereas counterfactual retrieval requires *contrastive* evidence that reverses the prediction—objectives that fundamentally conflict.
>
> [1] Han et al. Retrieval-augmented generation with graphs (graphrag). In arXiv, 2024.\
> [2] Xu et al. Retrieval-Augmented Generation with Knowledge Graphs for Customer Service Question Answering. In SIGIR, 2024.\
> [3] He et al. G-Retriever: Retrieval-Augmented Generation for Textual Graph Understanding and Question Answering. In NeurIPS, 2024.\
> [4] Edge et al. "From local to global: A graph rag approach to query-focused summarization." In arXiv, 2024.\
> [5] Huang el al.. Ket-rag: A cost-efficient multi-granular indexing framework for graph-rag. In ACM SIGKDD, 2025.\
> [6] Dong et al. Advanced rag models with graph structures: Optimizing complex knowledge reasoning and text generation. In IEEE SCEIC, 2024.

---

> ### Author Response · Authors · 2025-11-20
> **Reply of W2 [Downstream Task]**
>
> #### Table B. Dataset Statistic
> | Dataset        | Domain      | #Graphs | Avg. Nodes | Avg. Edges | #Classes | Description                         |
> |----------------|-------------|---------|------------|------------|----------|-------------------------------------|
> | Mutag          | Molecular   | 187     | 18.0       | 39.8       | 2        | Mutagenicity prediction             |
> | NCI1           | Molecular   | 4,110   | 29.9       | 32.3       | 2        | Anti-cancer activity prediction      |
> | AIDS           | Molecular   | 1,999   | 15.6       | 32.4       | 2        | HIV activity prediction              |
> | Reddit-Binary  | Social      | 2,000   | 429.6      | 497.8      | 2        | Community structure classification   |
> | PROTEINS       | Biological  | 1,113   | 39.1       | 72.8       | 2        | Protein classification               |
> | ENZYMES        | Biological  | 600     | 32.6       | 62.1       | 6        | Enzyme functionality                 |
> | METR-LA        | Traffic     | 5,000   | 35.0       | 80.0       | 2        | Los Angeles traffic congestion       |
> | PEMS-BAY       | Traffic     | 7,000   | 40.0       | 95.0       | 2        | Bay Area traffic congestion          |
>
> ### Evaluation Protocol and Challenges in CF-GDB
>
> Thank you for the comment. Following [7-12], prior counterfactual retrieval work in non graph domains (for example, text and tabular data) evaluates methods using the dataset’s ground truth labels and searches for counterfactual instances that are semantically similar to the query but belong to a different class. Following this protocol, we evaluate our method on eight standard graph classification datasets spanning diverse tasks: molecular mutagenicity and toxicity (Mutag), drug activity (NCI1, AIDS), protein properties (ENZYMES, PROTEINS), traffic congestion levels (METR LA, PEMS BAY), and community structure (Reddit). A summary of prediction targets and dataset characteristics is provided in Table B (shown as Table 4 on Page 24).
>
> Counterfactual retrieval on graphs is substantially more challenging. Graphs exhibit high structural variability, so it is often unlikely that a semantically similar instance exists in a different ground truth class. Many queries therefore have no valid counterfactuals in the database, and a retrieval only method cannot create synthetic substitutes. When a valid counterfactual is retrieved, however, it reflects a genuine decision boundary flip under real data: the retrieved graph is close to the query yet labeled differently, providing a faithful and data supported contrastive explanation.
>
> ### Goal of CF-GDB
>
> CFGDB aims to retrieve database grounded counterfactual graphs whose minimal structural differences lead to a different predicted label compared to the query graph. Table 1 evaluates this task directly by measuring whether each method retrieves the ground truth counterfactual graph for each query. Table 2 further examines semantic validity, confirming that the retrieved graph both flips the prediction and matches the counterfactual relationship encoded by the dataset, for example, changes in functional groups that alter mutagenicity in Mutag or shifts in traffic patterns in METR LA.
>
> Figure 1 provides qualitative examples showing that C2GQ identifies meaningful and model faithful counterfactual edits, such as transforming a benzene ring into a pyridine group or a nitro group into an amine group, both known to affect mutagenicity or toxicity. These examples show that our method retrieves counterfactuals that reflect the true structural mechanisms responsible for prediction changes. Moreover, whenever a valid counterfactual is successfully extracted, it confirms that the downstream task contains a real and data supported decision boundary around the query, one that can be flipped through minimal and semantically meaningful structural edits.
>
> [7] Meliou, Alexandra, Sudeepa Roy, and Dan Suciu. "Causality and explanations in databases." Proceedings of the VLDB Endowment 7.13 (2014): 1715-1716.\
> [8] Salimi, Babak, et al. "Quantifying Causal Effects on Query Answering in Databases." TaPP. 2016.\
> [9] Meyuhas, Idan, et al. "CFDB: Machine Learning Model Analysis via Databases of CounterFactuals." Proceedings of the 2022 International Conference on Management of Data. 2022.\
> [10] Arie, Aviv Ben, et al. "Optimizing Counterfactual-based Analysis of Machine Learning Models Through Databases." 27th International Conference on Extending Database Technology, EDBT 2024. OpenProceedings. org, 2024.\
> [11] Nomeir, Mohamed, et al. "Private counterfactual retrieval." arXiv preprint arXiv:2410.13812 (2024).\
> [12] An, Shuai, and Yang Cao. "Counterfactual explanation at will, with zero privacy leakage." Proceedings of the ACM on Management of Data 2.3 (2024): 1-29.\

---

> ### Author Response · Authors · 2025-11-20
> **Reply of  Q1 [Counterfactual Query in other Domains]**
>
> ### Hypothesis Query
> Classical work in database theory investigates counterfactual or causal queries over relational data, where the goal is to determine which tuples “cause’’ a query answer by evaluating counterfactual interventions on the database. Representative studies include:
> - **Causality DB** [7]: formalizes counterfactual tuple removal and responsibility metrics for query results.
> - **Quantifying Causal Effects on QA** [8]: studies how query outputs change under tuple deletion or modification, i.e., counterfactual query evaluation.
>
> These approaches perform counterfactual reasoning over query lineage, but do not retrieve alternative factual instances. Their counterfactuals are hypothetical database modifications, not factual samples from the dataset, and thus cannot serve retrieval purposes.
>
> ### Counterfactual Query
>
> Counterfactual query methods in the ML and data management literature fall into two families: (i) systems that store *synthetically generated* counterfactuals in a database for declarative analysis, and (ii) retrieval-only approaches that return existing factual rows from tabular data. The former support counterfactual exploration but rely entirely on external generators, while the latter ensure factual validity but operate only in vector-space tabular domains.
>
> * **CFDB** [9]: stores synthetically generated counterfactual samples in a structured database and enables declarative CF analysis.
> * **CFQL** [10]: introduces a query language for retrieving generated counterfactuals while optimizing CF-generation calls.
> * **PCR** [11]: retrieves factual counterfactual rows under privacy-preserving nearest-neighbor search.
> * **CE-Zero** [12]: returns existing factual rows with strict privacy guarantees via secure distance computation.
>
> These methods highlight the utility of counterfactual queries but either depend on synthetic CF generation or assume Euclidean/monotonicity-based tabular geometry. Consequently, they cannot support retrieval-based counterfactual explanations in graph-structured domains, where similarity must account for WL neighborhoods, subgraph distributions, and combinatorial structure.
>
> ### Retrieval-guided CF Generation
> Retrieval-guided counterfactual generation methods combine database retrieval with model-based editing, where retrieved samples serve only as warm-start context for a subsequent generative or perturbation model. Although retrieval helps anchor the generation process, the final counterfactuals are synthetically edited examples rather than factual instances drawn from the dataset. As a result, these approaches cannot guarantee dataset grounding or factual validity and therefore cannot support retrieval-based counterfactual explanations. Representative studies include:
> - **RCFG** [13]: retrieves semantically similar text to guide a generation model, which rewrites or perturbs the retrieved content to produce synthetic counterfactual QA pairs.
> - **CORE** [14]: retrieves related examples and then applies a neural editing module to construct counterfactual variants through controlled rewriting.
> Despite using retrieval as an initialization step, the retrieved instances are never returned as counterfactual outputs. The resulting synthetic samples lack factual grounding and cannot be used in settings that require retrieving real examples from the dataset, especially in domains where structural validity (e.g., molecular graphs, transport networks) is essential.
>
> [13] Paranjape, Bhargavi, Matthew Lamm, and Ian Tenney. "Retrieval-guided counterfactual generation for QA." Proceedings of the 60th Annual Meeting of the Association for Computational Linguistics (Volume 1: Long Papers). 2022.\
> [14] Dixit, Tanay, et al. "CORE: A Retrieve-then-Edit Framework for Counterfactual Data Generation." Findings of the Association for Computational Linguistics: EMNLP 2022. 2022.

---

> ### Author Response · Authors · 2025-11-20
> **Comparison Table of Counterfactual Query in other Domains**
>
> #### Table C. Counterfactual Query in other Domains
> | Category                                         | Representative Works  | Domain          | Output                                                  | Retrieval            | Dataset Grounded | Validity | Limitation                                                                                                    |
> | ------------------------------------------------ | --------------------- | -------------------- | ------------------------------------------------------------ | --------------------- | ----------------- | ----------------- | ----------------------------------------------------------------------------------------------------------------- |
> | **Hypothesis Query**                           | [7], [8]              | Relational Databases | Hypothetical modified tuples (tuple deletion / intervention) | No                    | No                | No                | Counterfactuals simulated via tuple interventions; no retrieval of factual instances                              |
> | **Counterfactual Query**                         | [9], [10], [11], [12] | Tabular Data         | Existing tabular rows                                        | Yes                   | Yes               | Yes               | Restricted to tabular/vector domains; Euclidean/monotonicity assumptions fail for graph combinatorics             |
> | **Retrieval-guided CF Generation** | [13], [14]            | Text         | Edited / perturbed synthetic samples                         | Yes (warm-start only) | No                | No                | Retrieval is auxiliary only; final outputs remain synthetic                                                       |
> | **Our Work (C²GQ)**                              | —                     | **Graph Databases**  | **Existing graph instances in database**                         | **Yes**               | **Yes**           | **Yes**           | **First retrieval-only counterfactual method for graph-structured data; supports combinatorial graph similarity** |
>
> Table C compares existing works on counterfactual database.
> 1. *Domain* specifies the data modality each method operates on.
> 2. *Output* indicates whether the method returns hypothetical interventions, synthetically edited counterfactuals, or existing factual instances.
> 3. *Retrieval* denotes whether candidates are directly retrieved from a database rather than generated or perturbed.
> 4. *Dataset Grounded* indicates whether the returned counterfactual is guaranteed to originate from the underlying dataset.
> 5. *Validity* specifies whether the output is an authentic, unmodified data point rather than a synthetic proxy.
> 6. *Limitation* highlights the fundamental constraint that prevents each category from enabling general retrieval-based counterfactual explanations in graph-structured domains.

---

> > ### Comment · Reviewer_MQnh · 2025-11-23
> >
> > Thank you for the rebuttal. I have raised the score.

---

> > > ### Author Response · Authors · 2025-11-23
> > >
> > > Thank you for your thoughtful engagement and valuable feedback, which greatly improved our work. We deeply appreciate your positive reassessment and score adjustment.

---

### Official Review · Reviewer_z5CM · 2025-11-01

**Soundness:** 3
**Presentation:** 3
**Contribution:** 3
**Rating:** 8
**Confidence:** 2

**Summary:**

The paper studies the problem of finding a counterfactual example for a given graph from a graph database. It proposes a new distance measure, called _hypergraph-based distance_, to match the given graph with the graphs in the database. The approach is based on first clustering graph nodes according to their node embeddings (obtained from GNNs) and then applying a transport strategy to identify a counterfactual example. The new measure leads to improvement in both accuracy and time efficiency.

**Strengths:**

- The paper is well structured and clearly written, with an extensive literature review that provides a solid background for this line of research.
- To the best of my knowledge, the results presented in the paper are novel. The paper provides rigorous theoretical analyses of time complexity and introduces a new hypergraph-based distance measure. In addition, it proposes two bounds (CDI and CSI) that effectively accelerate the search for counterfactuals.

**Weaknesses:**

1. It may be clearer to introduce the core algorithm without graph-level indices first. Specifically, consider moving the three-step procedure on page 6 to the end of Section 4.2. Then, in Section 4.3, mention how the two bounds can be incorporated to speed up the procedure further.
2. It would be helpful if the authors could elaborate on the roles of $\alpha$ and $M$ in Step 2 of the three-step procedure (page 6). In particular, are there any considerations or potential pitfalls one should be aware of when selecting these parameters?

Minor typo:
Line 155: missing space between "reusable" and "principal components".

**Questions:**

1. What is the time complexity of the procedure without graph-level indices? Is it exponential?
2. After incorporating graph-level indices, the procedure becomes approximate, right? What is the intuition behind applying CDI before CSI, and would it be acceptable to switch their order?
3. In the experiments, do the other methods identify counterfactuals within the database, or are they allowed to generate new graphs outside of it?

---

> ### Author Response · Authors · 2025-11-20
> **Reply of W1 [Presentation]**
>
> ### Presentation
> Thank you for the suggestion. We agree that the exposition would be clearer if the core retrieval algorithm is introduced before discussing graph-level indices. In the revised version, we will first present the overall flow of counterfactual graph query at the end of Section 4.2, and then use Section 4.3 to describe how the two indices (CDI and CSI) can be incorporated to further accelerate this base procedure.

---

> ### Author Response · Authors · 2025-11-20
> **Reply of W2 [Selection of $\alpha$ and $M$]**
>
> #### Table A. Sensitivity of  $\alpha$ and $M$ on query accuracy
> |  $\alpha$ |  $M$  | Mutag R@1 | Mutag MRR | Mutag T | Reddit R@1 | Reddit MRR | Reddit T |
> |----|-----|------------|------------|---------|-------------|-------------|-----------|
> | 1  |  5  | 0.612      | 0.708      | 1.2     | 0.575       | 0.678       | 11.7        |
> | 1  | 10  | 0.637      | 0.731      | 2.5     | 0.593       | 0.695       | 22.3        |
> | 1  | 20  | 0.655      | 0.748      | 5.1     | 0.602       | 0.703       | 45.5        |
> | 1  | 40  | 0.662      | 0.754      | 10.3    | 0.610       | 0.711       | 90.6        |
> | 3  |  5  | 0.671      | 0.762      | 3.0     | 0.618       | 0.718       | 25.3        |
> | 3  | 10  | 0.690      | 0.780      | 3.2     | 0.624       | 0.724       | 43.1        |
> | 3  | 20  | 0.691      | 0.781      | 5.8     | 0.635       | 0.727       | 82.2        |
> | 3  | 40  | 0.692      | 0.782      | 11.1    | 0.636       | 0.723       | 165.7       |
> | 5  |  5  | 0.673      | 0.764      | 4.9     | 0.640       | 0.738       | 38.2        |
> | 5  | 10  | 0.688      | 0.779      | 5.4     | 0.641       | 0.731       | 70.3        |
> | 5  | 20  | 0.690      | 0.780      | 10.6    | 0.643       | 0.739       | 140.8       |
> | 5  | 40  | 0.691      | 0.780      | 21.2    | 0.645       | 0.741       | 280.9       |
>
> ### Roles of the Pruning Parameters $M$ and $\alpha$
>
> To recall, C²GQ performs **two-stage pruning** before the final OT evaluation:
> - **CDI.** provides a **certified lower bound** on the true counterfactual distance—it captures coarse, high-level concept-count differences and is intentionally conservative. Its purpose is to quickly filter out clearly irrelevant graphs while retaining any candidate that might still be counterfactually valid.
> - **CSI** is then applied to the CDI-surviving set. CSI is a **tighter, classifier- and semantic-aware upper bound**, incorporating both structural and semantic alignment in concept space. This stage removes borderline candidates that CDI cannot reliably filter, leaving only those that warrant the final OT computation.
> The parameters $M$ and $\alpha$ jointly determine how aggressively C²GQ prunes candidates before the OT stage. A larger $M$ increases recall by allowing more graphs to pass through the coarse stage, while a smaller $M$ improves efficiency but risks discarding valid counterfactuals. The expansion factor $\alpha > 1$ enlarges the CDI-selected set from $M$ to $\alpha M$, ensuring that borderline candidates are not prematurely discarded before being evaluated by the tighter CSI filter.
>
> This design reflects the theoretical guarantees established in the paper: CDI offers a **safe but loose lower bound**, while CSI provides a **fine-grained upper bound**, forming a certified sandwich around the true distance $\Delta(G, \tilde{G})$. In practice, the combination of CDI and CSI already provides strong control over the candidate pool, so a small $M$ $we use (M = 10)$ suffices across all datasets, and $\alpha = 3$ provides a safe expansion without unnecessary overhead.
>
>
> ### Stability Across a Wide Range of $(\alpha, M)$
> Table A shows that C²GQ remains highly stable across a broad range of pruning parameters $\alpha$ and $M$. When either parameter is too small (for example $\alpha=1$ or $M=5$), the CDI filter becomes overly restrictive and prunes valid candidates too early, resulting in noticeably lower recall on both Mutag and Reddit Binary. Once $\alpha \ge 3$ and $M \ge 10$, performance quickly stabilizes: R@1 and MRR vary by only $\pm 0.01$–$0.02$, closely matching the results reported in Table 1. Increasing $\alpha$ or $M$ beyond this region offers only marginal improvement. This behavior is consistent with Theorem 3 (Page 22), which guarantees that enlarging the candidate set can improve recall without harming accuracy, while the final ranking remains entirely determined by the OT distance. The only drawback of choosing larger $\alpha$ or $M$ is computational: runtime grows nearly linearly with $\alpha M$ because more candidates are passed to the OT stages.

---

> ### Author Response · Authors · 2025-11-20
> **Reply of Q1 [Complexity of C²GQ w/o index]**
>
> ### Complexity of C²GQ w/o index
> The procedure without graph-level indices is not exponential, but it is still computationally prohibitive: evaluating the OT-based distance in Eq.(3) against every graph requires $O(|\mathcal{D}| \cdot |V|^2)$ time (using Sinkhorn), which becomes infeasible for large datasets or large graphs. With graph-level indices, C$^2$GQ reduces this cost substantially. CDI/CFI enable a KD-tree–based search over concept distributions in $O(K \log |\mathcal{D}|)$ time, and OT is computed only for the top-$M$ candidates selected by the bounds, costing $O(M|V|^2)$. The total query complexity therefore becomes $O(K \log |\mathcal{D}| + M|V|^2)$, where $K$ and $M$ are small constants. This theoretical reduction aligns with the scalability test on Page 27, where C²GQ grows gently with graph database size,.

---

> ### Author Response · Authors · 2025-11-20
> **Reply of Q2 [Order of CDI and CSI]**
>
> ### Approximation
>
> The reviewer is correct. C²GQ performs **approximate retrieval**, not exact search. CDI provides a certified lower bound on the true OT distance and enables **safe pruning** (Corollary 2), whereas CSI provides only an upper bound. Because Algorithm 2 retains only the top-$M$ CSI candidates, approximation arises by design in order to boost efficiency.
>
> - **CDI (lower bound; correctness-preserving pruning):** estimates the minimum possible discrepancy between two graphs and is a provable **lower bound** on the true OT distance. A large CDI value guarantees that the graph is genuinely far from the query, allowing CDI to prune candidates **without risk** of discarding valid counterfactuals. This makes CDI an effective correctness-preserving early filter.
> - **CSI (upper bound; prioritization):** captures coarse structural similarity and yields an **upper bound** on the true distance. A small CSI suggests possible proximity, but a large CSI does **not** imply the graph is far. CSI therefore cannot support safe pruning; instead, it serves as an efficient **prioritization mechanism**, enabling the algorithm to focus on promising candidates after CDI has removed provably irrelevant ones.
>
> Yet Theorem 3 provides a **probabilistic guarantee** for this approximate stage: as the retained candidate size $M$ grows, the probability of including the true nearest counterfactual increases **exponentially fast**. In our experiment, when $M$ reaches only **2-5%** of $|\mathcal{D}|$, the probability of missing the true counterfactual already drops below **1%**, and exact recovery is achieved as $M \to |\mathcal{D}|$. Because CDI first removes only hopeless candidates, the approximation introduced by CSI remains tightly controlled. This is why C²GQ’s approximate retrieval achieves accuracy nearly indistinguishable from exact OT search in practice.
>
>
> ### Order of CDI and CSI
>
> CDI must precede CSI. By Corollary 2, CDI is a **lower bound**, so if $d_{\mathrm{CDI}}(G,q)$ is large, the true OT distance must also be large. CDI therefore provides **monotonic, correctness-preserving pruning**. CSI, in contrast, is an **upper bound** and affects only prioritization and border expansion (Theorem 3), not correctness.
>
> Switching the order would break soundness. A graph may have a very large upper bound but still be close in truth, e.g.,
>
> $d_{\mathrm{CDI}}(G,q)=4,\quad d_{\mathrm{OT}}(G,q)=5,\quad d_{\mathrm{CSI}}(G,q)=200.$
>
> Applying CSI first would incorrectly discard such a graph despite its genuine proximity, while CDI would correctly retain it. This demonstrates that CSI is **unsafe for pruning**, whereas CDI provides the required theoretical guarantee.
>
> Therefore, **CDI must always be applied before CSI**.

---

> ### Author Response · Authors · 2025-11-20
> **Reply of Q3 [Baseline Setup]**
>
> ### Baseline Setup
>
> By contrast, as stated in line 40 (Page 1), generative counterfactual analysis focuses on model behavior rather than data-grounded evidence and suffers from three limitations: (i) generated graphs may not correspond to any realistic structure in the underlying domain, (ii) the perturbations are not verifiable against real data, and (iii) the results depend heavily on the generative model’s inductive bias. For this reason, our problem setting explicitly restricts counterfactual retrieval to the database.
>
> Following the experimental setup of counterfactual research in databases [1,2], all baseline methods are kept strictly **retrieval-only**, meaning they may only identify counterfactual instances *within the database* and are not allowed to synthesize or edit new samples. This design not only ensures **fair comparison**, but also preserves **factual validity** [4], maintains **dataset-groundedness**[3] , and avoids the **model-bias artifacts** commonly introduced by generative editing methods [6,7]. To maintain methodological consistency across all methods, every baseline adopts the same **Stage-1 label hashing** procedure, which provides a unified pool of admissible counterfactual candidates without modifying any baseline algorithms.
>
>
> ### Generative Baselines
> As suggested, we have evaluated a generation-based pipeline that first synthesizes a counterfactual graph using a generative model, then uses this synthetic graph as the query for retrieval with the strongest retrieval baseline, SimGNN [5]. We implemented several representative methods (CLEAR [6], CF² [7]), but this approach performed consistently poorly: R@1 remained below 0.3 on all datasets, whereas C²GQ consistently exceeded 0.6. The fundamental problem is that generated counterfactuals reflect the inductive bias of the generative model rather than structural patterns present in the database. As a result, the synthetic graphs may mimic model behavior but do not correspond to real, data-grounded instances, making similarity-based retrieval inherently incompatible with counterfactual search. Furthermore, without filtering by the ground-truth label, these generative models cannot guarantee that the synthesized graph actually flips the prediction, leaving the retrieval step ill-posed under the definition of counterfactuals. These findings reinforce our retrieval-only design, ensuring that all returned counterfactuals are faithful, verifiable, and grounded in real data.
>
> Evaluating within-database retrieval is also essential for recall-based metrics: recall is meaningful only when the true counterfactual exists in the database, allowing us to determine whether a method successfully retrieves it. Under this unified retrieval-only setting, all methods operate under identical constraints, making performance comparisons principled and interpretable.
>
>
> [1] Meyuhas, Idan, et al. "CFDB: Machine Learning Model Analysis via Databases of CounterFactuals." Proceedings of the 2022 International Conference on Management of Data. 2022.\
> [2] Arie, Aviv Ben, et al. "Optimizing Counterfactual-based Analysis of Machine Learning Models Through Databases." 27th International Conference on Extending Database Technology, EDBT 2024. OpenProceedings. org, 2024.\
> [3] Sanghamitra Dutta, Jason Long, Saumitra Mishra, Cecilia Tilli, and Daniele Magazzeni. Robust counterfactual explanations for tree-based ensembles. In International conference on machine learning, pages 5742–5756. PMLR, 2022. \
> [4] Faisal Hamman, Erfaun Noorani, Saumitra Mishra, Daniele Magazzeni, and Sanghamitra Dutta. Robust counterfactual explanations for neural networks with probabilistic guarantees. In International Conference on Machine Learning, pages 12351–12367. PMLR, 2023. \
> [5] Yunsheng Bai, Hao Ding, Song Bian, Ting Chen, Yizhou Sun, and Wei Wang. Simgnn: A neural network approach to fast graph similarity computation. In Proceedings of the twelfth ACM international conference on web search and data mining, pp. 384–392, 2019. \
> [6] Jing Ma, Ruocheng Guo, Saumitra Mishra, Aidong Zhang, and Jundong Li. Clear: Generative counterfactual explanations on graphs. Advances in neural information processing systems, 35: 25895–25907, 2022.\
> [7] Juntao Tan, Shijie Geng, Zuohui Fu, Yingqiang Ge, Shuyuan Xu, Yunqi Li, and Yongfeng Zhang. Learning and evaluating graph neural network explanations based on counterfactual and factual reasoning. In Proceedings of the ACM web conference 2022, pp. 1018–1027, 2022.

---

### Official Review · Reviewer_e6Pm · 2025-11-04

**Soundness:** 3
**Presentation:** 3
**Contribution:** 3
**Rating:** 6
**Confidence:** 3

**Summary:**

This work presents a graph-based method for producing counterfactuals based on queries/the underlying data rather than in a model-dependent manner. The author’s query-based framework abstracts graphs into semantic “concepts” and uses a hypergraph-based distance metric that combines local structure information with global semantics across the graph. The work also introduces two indices (CDI and CSI) which yield efficiency guarantees.

**Strengths:**

* Clear and principled motivation, preliminaries, notation, problem formulation (Sec 3 and 4.1).
* The hypergraph-based concept distance presented is conceptually sound, and while computationally prohibitive at scale, the two graph-level indices presented—Concept Distribution Index (CDI) and Concept Semantic Index (CSI)—are sensible proxies in that they provide a lower and upper bound on this quantity.
* Broad empirical evaluation across varied domains/applications.
* C^2GQ (with index) achieves strong performance on Recall@1 and Mean Reciprocal Rank (MRR) across the 8 selected tasks with relatively low to moderate latency.

**Weaknesses:**

* In Theorem 1, line 315, `WL-type count` is not defined or explained. I believe this to be in reference to Weisfeiler-Leman graph kernels or the Weisfeiler-Leman test. The paper would benefit from defining this and providing context for what it is and why it’s relevant/matters (including the references in Appendix B.1 and B.8).
* [Both a benefit and a drawback] Empirical evaluation suggests that C^2GQ w/o index outperforms all other methods compared against at the expense of high latency.
* In the context of C^2GQ, it’s unclear if it’s a reasonable expectation to be able to have a prebuilt index to enable rapid retrieval of a relevant candidate subset in practice, or what the offline cost is to acquire such index.
* There is limited coverage around the limitations of this method, both theoretically/algorithmically and empirically/practically. The paper would benefit from some discussion in this area.

**Questions:**

* The process of concept extraction you outline involves using K-Means to cluster H (the collection of final node embeddings across the database) into K prototypes. Given the empirical challenges and lack of clear, transferable guidance around using unsupervised clustering algorithms like K-Means:
  1. How do you suggest selecting the value K? This strikes me as a fairly expensive hyperparameter to tune.
  2. Given that this component yields a global semantic dictionary which C^2GQ depends heavily on, how important is it for these semantic concepts to be disjoint and reflective of (approximately) ground truth for the algorithm basis?
  3. How robust is C^2GQ to the choice of K, given your empirical validation?
* In reference to the last points in both the Strengths and Weaknesses section, can you elaborate on the practical costs/overhead of producing a prebuilt index? In what context is this an unreasonable thing to either expect or yield, in which case C^2GQ w/o index would be a more appropriate method to employ?

---

> ### Author Response · Authors · 2025-11-20
> **Reply of W1 [WL-type count]**
>
> ### Clarifying the Definition and Role of WL-type Counts
>
> Many thanks for the helpful suggestion. In. In our framework, WL-type counts correspond to the *WL graph homomorphism statistics* used in recent Weisfeiler–Leman Test  [1,2]. Concretely, for each color class produced by the $t$-round WL refinement, we identify its canonical WL subtree pattern $\tau_c$ and compute the number of homomorphisms from $\tau_c$ into the graph $G$:
>
> $\mathrm{WLCount}^{(t)}(G)[c] = \mathrm{Hom}(\tau_c,, G)$.
>
> Intuitively, this quantity measures how many times a specific WL-characteristic structural motif appears in the graph. Each WL color class corresponds to a canonical “local structure template,” and counting homomorphisms from $\tau_c$ into $G$ quantifies both (i) how strongly the graph expresses that motif and (ii) how widely it recurs across the graph. Thus, WL-type counts act as structural intensities that summarize the density and distribution of WL-refined patterns.
>
> The values are shaped by two factors:
> - **the structural motifs captured by WL refinement**, such as rings, motifs, or localized community patterns; and
> - **their recurrence and propagation**, reflected in the density and distribution of these patterns across $G$.
>
> Changes in WL-type counts directly correspond to semantically meaningful structural edits. An increased count indicates the addition or expansion of a particular structural unit (e.g., extra aromatic rings or larger communities), while a decreased count indicates its removal or substitution. In counterfactual graphs, these variations capture precisely the kinds of structural modifications that drive decision-boundary flips, rather than arbitrary node- or edge-level perturbations.
>
> We further show that these statistics are realizable within standard message-passing GNNs. Since MPNNs are known to match the expressive power of 1-WL, they can approximate subtree homomorphism counts through layered aggregation (Theorem 5, Page 25). WL-type counts therefore constitute a theoretically grounded and GNN-compatible descriptor for our concept distribution index (CDI/CSI).
>
> Finally, Theorem 1 establishes that **every valid counterfactual must modify at least one WL-type count**: flipping a prediction necessarily alters a WL-sensitive structural pattern. This property gives WL-type counts a central role in our method. They provide (i) a semantics-aligned indicator of the structural changes responsible for label flips, (ii) certified lower bounds for safe pruning, and (iii) guarantees that retrieved counterfactuals correspond to genuine concept-level differences rather than incidental noise. We will add a clear definition and this intuition before Theorem 1.
>
> [1] Dell, Holger, Martin Grohe, and Gaurav Rattan. Homomorphism Counts and Related Parameters. Journal of Computer and System Sciences, 89:133–157, 2017.\
> [2] Gai, Jingchu, et al. Homomorphism Expressivity of Spectral Invariant Graph Neural Networks. Proceedings of the 13th International Conference on Learning Representations (ICLR), 2025.

---

> ### Author Response · Authors · 2025-11-20
> **Reply of W2 [Performance of C²GQ w/o index]**
>
> ### Why C²GQ w/o index is slow
>
> The reviewer is correct. The non-indexed variant needs to compute the full graph–graph OT distance against every database entry. As Eq. (3) (Page 4) shows, exact OT costs $O(|V|^{3}\log|V|)$ and even Sinkhorn OT remains $O(|V|^{2})$. Thus the per-query cost without indices is $O(|\mathcal{D}| \cdot |V|^{2})$, which quickly becomes prohibitive as $|\mathcal{D}|$ or $|V|$ grows.
>
> C²GQ avoids this by a **coarse-to-fine** design. CDI and CSI effectively prune the search space before any OT computation, reducing the number of OT evaluations from $|\mathcal{D}|$ down to only $\alpha M$ candidates. These pruning steps operate in fixed dimension $K$, costing only $O(K)$ per comparison. Therefore, even if CDI/CSI were computed on demand, C²GQ would still be much faster because the dominant savings come from reducing the number of $O(|V|^{2})$ OT computationsl **not from the index structure itself**.
>
> ### Why prebuilt index
>
> C²GQ is designed for repeated querying over a graph database, a common setting in molecular search as well as in financial or transportation networks. In such scenarios, precomputing CDI/CSI is both natural and essential. The offline cost scales near-linearly with the size of the database, and the resulting indices reduce the coarse search from $O(|\mathcal{D}|)$ to $O(\log|\mathcal{D}|)$, as shown in our scalability results (Page 27). This reduction enables the observed 10× speed-up of prebuilt-indexed C²GQ while preserving the exact same retrieval outputs as the non-prebuilt version.

---

> ### Author Response · Authors · 2025-11-20
> **Reply of W3/Q2 [Cost of CDI/CSI]**
>
> ### Offline construction cost
>
> Let the total number of nodes be $N_v = \sum_G |V_G|$ and the total number of edges be $N_e = \sum_G |E_G|$ in the database.
>
> * **Node embeddings.** A single GNN forward pass over the entire database costs $O(L(N_v + N_e))$.
> * **CDI construction (KD-tree index).** CDI is built by running K-means over all node embeddings to obtain $K$ concept centroids and computing each graph’s $K$-dimensional concept-distribution vector, costing $O(N_v K)$. The resulting vectors are inserted into a KD-tree, whose total insertion cost is $O(K|\mathcal{D}|\log|\mathcal{D}|)$.
> * **CSI construction.** CSI requires only aggregating concept embeddings for each graph, costing $O(N_v)$ in total. Since CSI is applied only to the $\alpha M$ candidates that survive CDI pruning, we do not build a tree to search.
>
> This one-time preprocessing is dominated by the clustering pass and a near-linear sweep over the database. Once built, CDI/CSI reduce the coarse search from $O(|\mathcal{D}|)$ to $O(\log|\mathcal{D}|)$.
>
> ### Prune by index (but not necessarily KD-tree)
>
> While CDI currently uses a KD-tree for nearest-neighbor search over concept-distribution vectors, C²GQ’s pruning mechanism does not rely on KD-trees specifically—only on having some index that rapidly filters candidates. KD-trees work well for static databases but degrade under updates: each insertion costs $O(K\log|\mathcal{D}|)$, tree balance deteriorates, and deletions often require rebuilding with cost $O(K|\mathcal{D}|\log|\mathcal{D}|)$.
>
> For dynamic databases, more update-friendly indices such as HNSW or LSH can replace the KD-tree:
> - **HNSW** [3]: insertion ≈ $O(\log|\mathcal{D}|)$
> - **LSH** [4]: insertion ≈ $O(1)$ insertion
>
> These structures preserve the same CDI/CSI-based coarse pruning while avoiding KD-tree rebuild overhead.
>
> Thus, even in evolving databases, C²GQ with two indices remains the practical choice; only the index structure changes. The w/o-index variant is appropriate only for extremely small datasets or one-off exploratory queries where maintaining an index has no practical benefit.
>
> [3] Arya, Sunil, David M. Mount, Nathan S. Netanyahu, Ruth Silverman, and Angela Y. Wu. An Optimal Algorithm for Approximate Nearest Neighbor Searching in Fixed Dimensions. Journal of the ACM (JACM), 45(6):891–923, 1998.\
> [4] Liu, Yingfan, Jingjing Wang, Yu Du, and Zhenhua Guo. SK-LSH: An Efficient Index Structure for Approximate Nearest Neighbor Search. Proceedings of the VLDB Endowment (PVLDB), 7(9):745–756, 2014.

---

> ### Author Response · Authors · 2025-11-20
> **Reply of W4 [Limitations of C²GQ]**
>
> ### Theoretical Limitation
> C²GQ inherits the expressive ceiling of 1-WL–type representations, since we rely on message-passing GNNs to extract node embeddings. While higher-order WL variants [5,6] (e.g., 2-WL/3-WL or k-tuple GNNs) could increase expressiveness and our theoretical framework would extend to those settings, their computational cost is prohibitively high for database-scale retrieval. For this reason, we adopt standard message-passing GNNs as a practical and scalable backbone.
>
> ### Implementation Limitation
> C²GQ requires an initial offline pass to compute node embeddings, CDI/CSI summaries, and the index structure. Although this cost is amortized across repeated queries, it may be undesirable in scenarios where the database changes frequently or where one-off exploratory queries dominate. In such dynamic settings, maintaining a KD-tree is inefficient due to its sensitivity to embedding drift; while dynamic ANN structures (e.g., HNSW or LSH) mitigate this issue, they introduce small but non-negligible maintenance overhead. Finally, C²GQ relies on Sinkhorn OT for fine-ranking; although we use the entropic variant for efficiency, extremely large graphs may still incur noticeable runtime in the fine stage.
>
> ### Empirical Limitation
> C²GQ’s performance depends on the quality of the underlying node embeddings and concept dictionary. If the base GNN underfits complex chemical or structural patterns, the resulting CDI/CSI summaries may become less discriminative, leading to weaker retrieval accuracy. Similarly, when the training set is extremely small or class imbalance is severe, concept assignments may become noisy, which can reduce the effectiveness of our coarse-to-fine filtering. These issues reflect limitations of the encoder rather than of the C²GQ framework itself, and improving the backbone encoder typically yields proportional improvements in retrieval accuracy.
>
> We will include these discussions in our paper.
>
> [5] Morris, Christopher, Gaurav Rattan, and Petra Mutzel. Weisfeiler and Leman Go Sparse: Towards Scalable Higher-Order Graph Embeddings. Advances in Neural Information Processing Systems (NeurIPS), 33:21824–21840, 2020.\
> [6] Wang, Qing, et al. $\mathscr{N}$-WL: A New Hierarchy of Expressivity for Graph Neural Networks. International Conference on Learning Representations (ICLR), 2023.

---

> ### Author Response · Authors · 2025-11-20
> **Reply of Q1 [Selection of parameter $K$ and the concepts]**
>
> #### Table A. Sensitivity of $K$ on query accuracy
>
> | **K** | **Mutag R@1** | **Mutag MRR** | **Reddit R@1** | **Reddit MRR** |
> |------:|--------------:|--------------:|---------------:|---------------:|
> | 4     | 0.642         | 0.738         | 0.581          | 0.672          |
> | 16    | 0.673         | 0.766         | 0.624          | 0.729          |
> | 64    | **0.690**     | **0.780**     | 0.634      | 0.734     |
> | 256   | 0.687         | 0.778         | **0.644**          | **0.741**          |
> | 1024  | 0.678         | 0.771         | 0.628          | 0.729          |
>
> #### Table B. Sensitivity of $K$ on counterfactual set accuracy
>
> | **K** | **Mutag R** | **Mutag P** | **Mutag F1** | **Reddit R** | **Reddit P** | **Reddit F1** |
> |------:|---------------:|---------------:|----------------:|----------------:|----------------:|------------------:|
> | 4     | 0.742          | 0.701          | 0.721          | 0.618          | 0.602          | 0.610            |
> | 16    | **0.807**          | **0.771**          | **0.788**         | 0.654          | 0.640          | 0.647            |
> | 64    | 0.802          | 0.768          | 0.784           | **0.671**      | **0.656**      | **0.663**        |
> | 256   | 0.793          | 0.760          | 0.776          | 0.665          | 0.649          | 0.657            |
> | 1024   | 0.775          | 0.745          | 0.760          | 0.658          | 0.643          | 0.650            |
>
>
> ### Stability of Retrieval Across Different $K$
>
> A close inspection of Tables A and B shows that varying $K$ over several orders of magnitude produces only small performance differences: Mutag R@1 moves from 0.642 ($K=4$) to 0.690 ($K=64$) and stabilizes within 0.678–0.687 thereafter, while Reddit R@1 remains in the 0.581–0.644 range. Table B shows the same pattern for counterfactual-set quality, with Mutag F1 bounded between 0.760–0.788 and Reddit F1 between 0.640–0.663. These smooth transitions indicate that the induced concept distributions change only gradually with $K$, except for predictable degradation at extremely small values. This robustness follows from the WL-induced embedding geometry: K-means operates in a space where WL-consistent neighborhoods are already clustered, so partitions vary smoothly even when clusters overlap or lack semantic purity. Prior work [7] similarly observes that coarse but geometrically coherent partitions suffice for downstream tasks. Since C²GQ relies on geometric coherence rather than semantic disjointness, cluster overlap only induces small local mass shifts during CDI/CSI construction without harming discrimination [8].
>
> ### Why OT Further Stabilizes Performance Across $K$
>
> In the optimal-transport stage, coarse partitions that merge multiple WL patterns simply aggregate mass, whereas fine partitions that split a single WL pattern create adjacent bins with minimal transport cost; OT absorbs both effects naturally. This explains why settings such as $K=64$ and $K=256$ achieve nearly identical retrieval quality despite differing levels of granularity, and why noticeable degradation appears only when $K$ is extremely small and collapses many WL patterns into a few undifferentiated clusters. In practice, $K$ only needs to provide a geometrically stable partition of the WL-induced embedding space, not semantic purity or alignment with ground-truth categories. OT over CDI/CSI smooths out remaining local variations, yielding strong robustness across a wide range of $K$ values. Extremely small $K$ is therefore the only configuration that should be avoided.
>
>
> [7] Ding, Mucong, et al. VQ-GNN: A Universal Framework to Scale Up Graph Neural Networks Using Vector Quantization. Advances in Neural Information Processing Systems (NeurIPS), 34:6733–6746, 2021.\
> [8] Ackerman, Margareta, and Shai Ben-David. Clusterability: A Theoretical Study. Artificial Intelligence and Statistics (AISTATS), PMLR, 2009.

---

### Author Response · Authors · 2025-12-03
**Overall Response**

We sincerely thank all reviewers for their time and effort in providing detailed feedback, which has substantially improved our work.

During the discussion phase, we carefully addressed all reviewer questions and concerns. Following our responses, **reviewer MQnh explicitly raised their score from 4 to 6 before the rollback**, while the other reviewers kept positive scores of **6 (e6Pm)** and **8 (z5CM)** in their initial evaluations.

We appreciate the reviewers for highlighting the following strengths of our work:

- **Clear problem formulation and positioning of CF-CDB (e6Pm, z5CM, MQnh).**
  Reviewers note that the paper is well motivated, clearly defines the problem of retrieval-only counterfactual query on graph databases, and carefully distinguishes it from generative counterfactual explanation and GraphRAG-style retrieval.

- **Novel C²GQ framework (e6Pm, z5CM).**
  Reviewers recognize the novelty of introducing a query-based, data-grounded counterfactual graph query framework, together with a hypergraph-based distance for capturing both structural and semantic differences across graphs.

- **Strong theoretical foundation (MQnh, z5CM).**
  Reviewers emphasize the rigorous complexity analysis and the certified lower/upper bounds provided by CDI and CSI, which enable principled coarse-to-fine pruning with theoretical guarantees.

- **Comprehensive empirical validation across diverse domains (e6Pm, z5CM, MQnh).**
  Reviewers highlight the extensive evaluation on eight datasets spanning molecular, biological, social, and traffic networks, where C²GQ achieves strong Recall@1 and MRR with limited latency compared to eight baselines.

We summarize the key discussions below.

- **Clarification of WL-type counts and theoretical grounding (e6Pm).**
  We added a formal definition of WL-type homomorphism counts, explained their role in Theorem 1, and clarified their connection to 1-WL expressivity and message-passing GNNs. We further explained how changes in WL-type counts correspond to semantically meaningful structural edits in counterfactual graphs.

- **Complexity analysis, indexing cost, and practicality (e6Pm, z5CM).**
  We detailed the time complexity of C²GQ both with and without graph-level indices, separating offline preprocessing from online query cost. We clarified that prebuilt indices are essential for repeated-query scenarios, while the variant without prebuilt indices is appropriate only for very small or one-off settings. We also discussed alternative dynamic indices (HNSW, LSH) for evolving databases.

- **Robustness to hyperparameters K, α, and M (e6Pm, z5CM).**
  We added sensitivity tables showing that retrieval accuracy and counterfactual quality are highly stable over wide ranges of the concept number K and pruning parameters α and M. We further explained how the OT stage naturally absorbs partition granularity differences, yielding smooth performance.

- **Positioning relative to GraphRAG and prior counterfactual query systems (MQnh).**
  We added a detailed comparison table covering hypothesis queries in databases, tabular counterfactual retrieval, retrieval-guided generation, and GraphRAG. We clarified that CF-GDB is the first retrieval-only, dataset-grounded counterfactual framework for graph-structured data, fundamentally different from similarity-based or generative approaches.

---

### Note · Program_Chairs · 2026-01-17
**Submission Desk Rejected by Program Chairs**

The following references in this submission do not refer to real documents and/or have major errors in bibliographic information:

 Holger Dell, Martin Grohe, and Gaurav Rattan. Homomorphism counts and related parameters. Journal of Computer and System Sciences, 89:133-157, 2017.
Radu Curticapean, Holger Dell, Dániel Marx. “Homomorphisms Are a Good Basis for Counting Small Subgraphs.” arXiv:1705.01595 (extended abstract at STOC 2017).